

# 3D high-resolution seismic imaging of the iron-oxide deposits in Ludvika (Sweden) using full-waveform inversion and reverse-time migration

Brij Singh[1], Michał Malinowski[1,2], Andrzej Górszczyk[1,3], Alireza Malehmir[4], Stefan Buske[5], Łukasz Sito[6]

and Paul Marsden[7]

[1]Institute of Geophysics, Polish Academy of Sciences Warsaw, 01-452, Warsaw, Poland
[2]Geological Survey of Finland, FI-02151, Espoo, Finland
[3]Institut des Sciences de la Terre (ISTerre), University Greoble Alpes, 38610, Grenoble, France
[4]Department of Earth Sciences, Uppsala University, 75236, Uppsala, Sweden
[5]Institute of Geophysics and Geoinformatics, TU Bergakademie Freiberg, D-09596, Freiberg, Germany
[6]Geopartner Sp. z o. o., 30-383, Kraków, Poland
[7]Nordic Iron Ore AB, 18291, Danderyd, Sweden

*Correspondence to*: Brij Singh (bsingh@igf.edu.pl)

**Abstract**

A sparse 3D seismic survey was acquired over the Blötberget iron-oxide deposits of the Ludvika Mines in south-central Sweden. The main aim of the survey was to delineate the deeper extension of the mineralisation and to better understand its 3D nature and associated fault systems for mine planning purposes. To obtain a high-quality seismic image in depth, we applied time-domain 3D acoustic full-waveform inversion (FWI) to build a high-resolution P-wave velocity model. This model was

subsequently used for pre-stack depth imaging with reverse time migration (RTM) to produce the complementary reflectivity section. We developed a data preprocessing workflow and inversion strategy for the successful implementation of FWI in the hardrock environment. We obtained a high-fidelity velocity model using FWI and assessed its robustness. We extensively tested and optimised the parameters associated with the RTM method for subsequent depth imaging using different velocity models: a constant velocity model, a model built using first-arrival traveltime tomography and a velocity model derived by

FWI. We compare our RTM results with *a priori* data available in the area. We conclude that, from all tested velocity models, the FWI velocity model in combination with the subsequent RTM step, provided the most focussed image of the mineralisation and we successfully mapped its 3D geometrical nature. In particular, a major reflector interpreted as a cross-cutting fault, which is restricting the deeper extension of the mineralisation with depth, and several other fault structures which were earlier not imaged were also delineated. We believe that a thorough analysis of the depth images derived with the combined FWI-

RTM approach that we presented here can provide more details which will help with better estimation of areas with high mineralization, better mine planning and safety measures.



## 1 Introduction

Application of reflection seismics has increased manifolds in the past decade for targets ranging from shallow to deep mineral deposits associated with the hardrock environment (see Malehmir et al., 2012 and references therein). The need for it has never been more urgent than now due to the fast depletion of shallower deposits and an exponential increase in demand for raw materials towards energy transition (Hofmann et al., 2018). The most significant feature that seismics bring is its ability to map geological features in deeper parts of the subsurface with much higher resolution than any other existing geophysical method such as electromagnetics or potential field methods as far as mineral exploration is concerned. The application of reflection seismic in mineral exploration has now matured, such that many successful 3D surveys have been conducted over the past three decades (Milkereit et al., 2000; Malehmir and Bellefleur, 2009; Cheraghi et al., 2012; White et al., 2012; Bellefleur et al., 2015; Koivisto et al., 2015; Yavuz et al., 2015; Bellefleur et al., 2019). Despite that, there is still a lot of hesitation towards the adoption of the seismic method as a standard tool for mineral exploration. Factors like low-impedance contrast between mineralisation and host rock, geological complexity, strong scattering of seismic waves, low signal-to-noise ratio (SNR), irregular shot and receiver geometries are some key challenges associated with the application of seismics in hardrock environment. Also, in a majority of cases, a standard time imaging workflow consisting of dip moveout (DMO) followed by post-stack time migration (PoSTM) is utilized. Unfortunately, this approach can fail to address all of the imaging challenges. Unlike the oil and gas exploration, where pre-stack depth migration (PreSDM) is often the standard imaging method, it has been only recently applied to characterize the geologically complex hardrock environment in a mineral exploration context (Hloušek et al., 2015; Singh et al., 2019; Heinonen et al., 2019; Bräunig et al., 2020; Brodic et al., 2021).

A major challenge in shifting from the aforementioned standard time-domain imaging to PreSDM is the non-availability of a robust velocity model building tool. Reflection tomography is usually employed to build the velocity model required for PreSDM, but the deficiency of coherent reflections typical for hardrock environment restricts its utilisation. Migration velocity analysis based on vertical velocity update and semblance are not valid for complex media (Al-Yahya, 1989). First-arrival traveltime tomography (FAT) had been successfully applied in many cases in the past for building velocity models in hardrock environment (Malehmir et al., 2018; Singh et al., 2019; Bräunig et al., 2020). However, since FAT only utilises first-arrival traveltime information, the resolution of the model is inherently limited. It also largely depends on the offset range being utilised for traveltime inversion which in terms of depth penetration generally limits to the first few tens or hundred meters from the surface – considering a small velocity gradient with depth of the underlying medium.

In recent decades, a new technique of velocity model building called full-waveform inversion (FWI) (Virieux and Operto, 2009; Tromp, 2020) has helped the hydrocarbon industry to solve complex imaging challenges, e.g. seeing through gas clouds and resolving shallow velocity heterogeneities. FWI brings unprecedented resolution in elastic/anelastic parameter models as compared to ray-based methods, however, it requires good-quality data, ideally with enhanced low frequencies and various recorded arrivals sampling the subsurface targets over a broad range of scattering angles. Usually, these conditions are hardly met by the seismic data acquired on land. Compared to marine datasets, seismic data acquired on land often suffers from low





SNR, strong elastic effects, large near-surface velocity contrasts, heterogeneous topography variations, etc. Nevertheless, a few successful case studies have been reported for 2D and 3D land datasets using acoustic/viscoacoustic FWI (Ravaut et al., 2004; Malinowski et al., 2011; Baeten et al., 2013; Adamczyk et al., 2014; Stopin et al., 2014; Cheng et al., 2017). But, to date, FWI in the mineral exploration context was almost exclusively focused on cross-hole/VSP data (Afanasiev et al., 2014).

In this work, we explore the potential of time-domain early-arrival acoustic FWI to build a high-resolution P-wave velocity
model for subsequent depth imaging using sparse 3D seismic data acquired over an iron-oxide mineralisation target at Ludvika (Central Sweden). Application of the early-arrival FWI is hampered in our case by the fact that due to the medium properties, first-arrivals are dominated by frequencies above 25 Hz. There is also a thin but heterogeneous weathering layer (Maries et al., 2017; Bräunig et al., 2020), as well as a small velocity gradient, which limits the penetration depth of refracted arrivals. Based on this Ludvika 3D dataset, we developed a data preprocessing workflow and a FWI strategy applicable to hardrock
seismic data for building a high-resolution velocity model. We also investigated the application of reverse-time migration (RTM) for subsequent depth imaging to produce high-quality depth images consistent with the FWI-derived velocity model, which may otherwise require some smoothing to be used in ray-based migrations (e.g., Kirchhoff PreSDM). According to our knowledge, this is the first application of the FWI-RTM imaging loop to a full 3D seismic survey acquired for mineral exploration in hardrock environment. Finally, we compare our imaging results with the available geological data to evaluate
improvements in the delineation of the mineralisation and fault zones.

This paper is organized as follows. In the 'Materials and methods' section, we describe the local geology, the data acquisition and the theoretical background of FWI and RTM. The 'Application to the Ludvika 3D dataset' section is divided into two parts. First, we describe all prerequisites required by FWI such as the starting model, data preprocessing, inversion parameters and strategy. This is followed by the FWI results and different approaches we used to quality-control (QC) them. The second
part of the section is dedicated to RTM and deals with data processing, computational aspects and finally the depth imaging results. In the 'Interpretation and discussion' section, we have discussed the results obtained from the present study and key findings for an improved seismic acquisition for future studies. Finally, we conclude our case study in the 'Conclusions' section.

## 2 Materials and methods

### 2.1 Geological background and earlier borehole and seismic studies

The Blötberget iron-oxide deposits at Ludvika are located within the Bergslagen mining district in south-central Sweden. For several centuries, the mining district had been central and famous for iron ore mining in Sweden. The Bergslagen mineral endowment is diverse and ranges from iron oxides to massive sulphides, skarns and is potentially rich in rare-earth elements (Rippa and Kübler, 2003; Stephens, 2009). The deposits occur within ca. 1.90-1.85 Ga felsic volcanic rocks surrounded by
migmatite and later granitic and pegmatitic intrusions (Fig. 1). The Blötberget mineralization is considered of 'apatite iron-



oxide type' or Kiruna-type with haematite and magnetite as the mineralization and 25 % - 60 % Fe content. The mineralisation occurs in three sheet-like bodies trending east-west namely: Kalygruvan, Hugget-Flygruvan and Sandellmalmen. Stratigraphically, the hematite rich zones (Hugget-Flygruvan) overlie the magnetic-rich zones (Kalygruvan). According to Nordic Iron Ore, the company which is currently operating the mine, mineralisation at Blötberget strikes in NE-SW direction

for several hundreds of metres and down to 800 m (based on drill hole data). The mineralisation thickness ranges between 10-50 m. In terms of structure, the mineralisation dips moderately (40°-50°) towards SE up to a depth of approximately 500 m, afterwards the dip becomes gentler in a listric-form manner (Malehmir et al., 2021; Markovic et al., 2020; Maries et al., 2017).

A detailed analysis of physical rock properties was also carried out based on several boreholes downhole logged in the area

(Maries et al., 2017). Downhole logging property measurements consisted of magnetic susceptibility, natural gamma radiation, formation resistivity, fluid temperature, fluid conductivity etc. Full-waveform sonic logging was also performed providing P- and S-wave velocities. The density of the core samples along the mineralisation was estimated in the lab. Magnetite and hematite mineralisation are characterised by the mean velocity and density of 5600 m/s and 4000 kg/m$^3$, respectively. Velocities in the host rock vary between 5100-6300 m/s, depending which rock types were intersected.


Prior to the acquisition of the sparse 3D seismic survey, a pilot 2D seismic study was conducted in the area with the aim of deep mineral targeting over the Blötberget mineralisation (Malehmir et al., 2017) along profile P1 marked in Figure 1. In addition to the standard time-domain imaging, an advanced Kirchhoff-based PreSDM was also applied to the 2D dataset, which showed the extent of the mineralisation clearly down to 1000 m depth (Bräunig et al., 2020). Later, RTM was also

applied along the same profile (Ding and Malehmir, 2021), which highlighted two sets of strong seismic reflectors dipping south-east which matched well with the known mineralisation. It also showcased two oppositely dipping reflectors intersecting the mineralisation and suggested the termination of extension of mineralisation further in depth.

## 2.2 Seismic data

In order to better understand the geometry of the deposits, as well as to better constrain structural features of the host rock, a

fixed-geometry 3D seismic survey was acquired in April-May 2019 within the frame of the H2020-funded Smart Exploration™ project. The acquisition covered a total area of about 3.8 km × 2 km (Fig. 1). The survey consisted of 1266 cabled (Sercel™ 428) and wireless receivers (Sercel Units and Wireless Seismic™ RT2) equipped with 10 and 28 Hz geophones. Receiver spacing was kept at 10 m uniformly throughout the survey except at some places where it was increased to 20 m to allow a larger survey area. The 32-ton Vibroseis source of TU Bergakademie Freiberg with 276 kN peak force and

a sweep frequency band of 10 – 160 Hz was used. Shot spacing was also kept at 10 m overlapping the receiver positions throughout the survey resulting in 1062 shot points in total. Shot points and receivers were mainly placed along the existing forest tracks with some receivers in the forest. The survey resulted in high-quality data with first breaks clearly visible up to a



full offset range of ~3.8 km. Details of the survey and some preliminary interpretation of the results, using conventional

processing workflows, can be found in Malehmir et al. (2021).


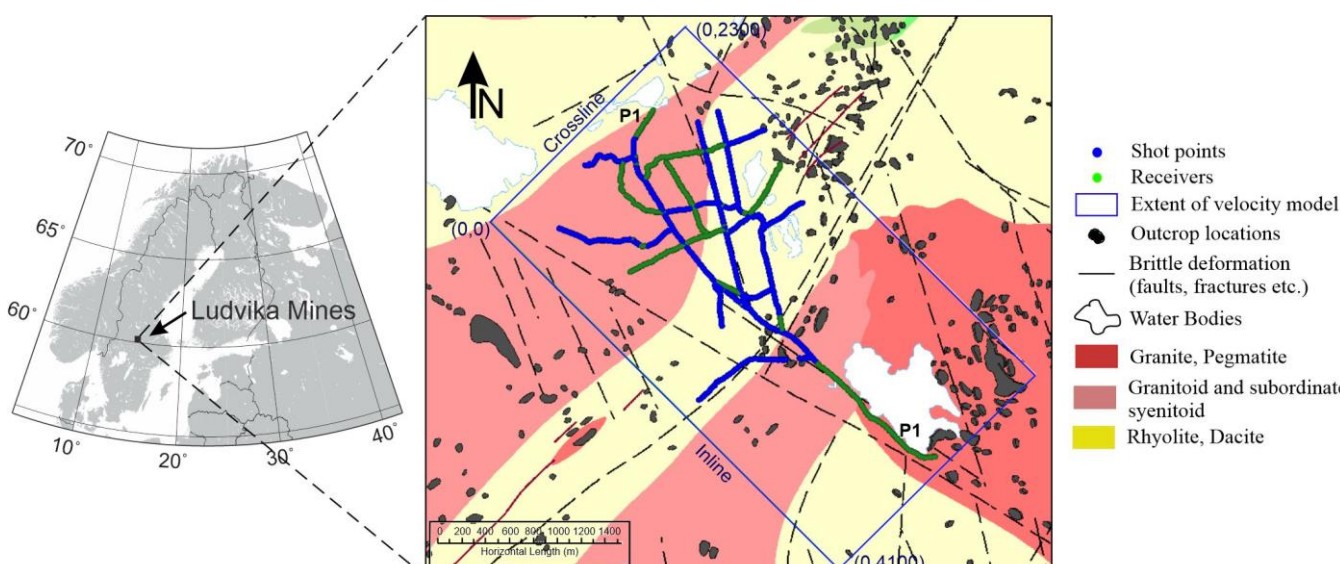

**Figure 1: 3D data acquisition geometry at Blötberget and bedrock geology (modified after Geological Survey of Sweden). Receivers of the sparse 3D survey are shown by green dots and shots by blue dots overlapping the receivers. The rectangular blue box shows the extent of velocity model space used for FWI and subsequent depth imaging using RTM in local coordinates.**

**2.3 Full-waveform inversion**

With an exponential increase in computational power in the last decades, FWI emerged as the preferred choice for high-resolution velocity model building due to its ability to utilise the entire information contained in the seismic trace. FWI can be either implemented in frequency or in the time domain, while the latter is usually used in 3D cases. In our approach, we used 3D time-domain viscoacoustic FWI implemented in the TOYXDAC_TIME code developed by the SEISCOPE consortium.

We used a finite-difference (FD) discretization of the acoustic FWI for forward formulation (see Hustedt et al., 2004 and references therein). The modelling engine is based on an explicit time-marching algorithm based on a staggered formulation of the first order velocity-stress wave equation. The time derivative is discretized by a second-order scheme while the spatial derivatives are discretized by the 4$^{th}$ order FD scheme. Sponge absorbing layers are implemented on the edges and sinc interpolation is used to localize source and receivers in the FD grid (Hicks, 2002).

The inversion scheme is based on the adjoint-formulation that uses the gradient of the misfit function to iteratively update the velocity models based on compliance formulation (Yang et al., 2016, 2018). The gradient is regularised with the Gaussian smoothing operator defined by its correlation lengths and the local wavelength. Different optimization schemes like steepest-descent (SD), L-BFGS, truncated Newton, etc. are implemented through the SEISCOPE Optimization Toolbox (Métivier and





Brossier, 2016), although in our case we mainly utilized preconditioned SD algorithm. An approximate Hessian is used as the
preconditioner in the optimization algorithm.

To increase the computational efficiency, TOYXDAC_TIME code is parallelised at two levels: the first level of parallelism is
built with Message Passing Interface (MPI), which tackles its own source i.e. one distributed memory MPI thread per shot
point. The second level is based on shared memory Open Multiprocessing (openMP). This level is based on the computation
of FD stencil loops and gradient loops per FWI iteration. In simpler terms, this allows the user to dedicate more cores per
source for a given node in an HPC system. This is helpful when the model space is large in size (i.e. in terms of the total
number of grid points) and memory storage is a key issue.

## 2.4 Reverse-time migration

Reverse time migration (RTM) belongs to the class of two-way wavefield extrapolation PreSDM methods. In recent times, it
has become a conventional choice for depth imaging in the case of complex media (such as subsalt imaging) thanks to an
increase in computational power (Zhou et al., 2018). Contrary to other imaging techniques, RTM is capable of using all types
of seismic phases that can be computed numerically. The unique advantage of this approach is that RTM is not based on
primary reflections only like in other existing methods which often mistakes non-primary waves as primary reflections, and
hence helps in reducing the migration artefacts to a great extent in cases where such secondary or multiple reflections occur
due to the complexity of the medium. In the latter case, RTM is able to accurately map the targeted features at their correct
locations compared to other PreSDM methods relying on first-arrivals only. For a complete overview of the history and
development of RTM, please refer to Zhou et al. (2018).

RTM aims to obtain accurate/angle-dependent estimation of reflection coefficients. The zero-lag cross-correlation imaging
condition for a single common source can be expressed as:

$$Image(x, y, z) = \sum_{t=0}^{T_{max}} S(x, y, z, t) R(x, y, z, t)$$

where, $(x,y,z)$ defines the spatial coordinates of the imaging point, $T_{max}$ is the maximum recording time and $S$ and $R$ represent
the source and receiver wavefields, respectively (Chattopadhyay and McMechan, 2008). Both, the receiver and source
wavefields, are independently propagated with the same scalar, two-way FD extrapolator. The receiver wavefield $R(x,z,t)$ is
backpropagated from the receiver location whereas the source wavefield $S(x,z,t)$ is propagated from the source location. The
image is obtained by cross-correlating the two wavefields at each time step (Claerbout, 1971). It is to be noted that the obtained
image is amplitude squared which means that image amplitude now has arbitrary scaling which ultimately depends on the
source strength, and so has no physical interpretation as reflection coefficient. This can be tackled by normalising the obtained



image amplitude by dividing the above equation by the square of source wavefield amplitudes $S^2(x,z,t)$. In this case, the source-normalized image will have the same (dimensionless) unit, scaling and sign as the reflection coefficient.

### 3 Application to the Ludvika 3D dataset

#### 3.1 Full-waveform inversion

#### 3.1.1 Starting model

The first step towards FWI is to have an initial velocity model that can predict the waveforms within half the dominant period for the data (Virieux and Operto, 2009). Usually, the starting velocity model for FWI is built by reflection tomography, but
due to the deficiency of coherent signals in hardrock seismic data, the method is certainly out of question. FAT has proven to be successful in few past case studies done in the hardrock environment, therefore we decided to use FAT for building the starting velocity model (Singh et al., 2019; Bräunig et al., 2020). Approximately 1.1 million traces were semi-automatically picked and manually corrected. We performed FAT using the inversion framework of Zhang and Toksöz (1998) implemented in the Geotomo TomoPlus software. We used all the shots and receivers from the 3D survey to build the starting velocity
model. The grid spacing for forward modelling was kept at $10 \times 10 \times 5$ m while the inversion was performed with a grid spacing of $20 \times 20 \times 5$ m (Fig. 2). A root-mean-squared (RMS) value of approximately 5 ms was obtained in 10 iterations. The FAT velocity model was resampled to a 10 m grid size as final output ($411 \times 231 \times 151$ cells). The upper boundary of the velocity model is 250 m above sea level. All the velocity models and subsequent depth images shown later in this article have the same configuration. A highly variable near-surface is observed in the NE direction of the model due to the presence of an
old tailing dam (Fig. 2a). Overall high velocities can be observed in the shallower part of the model with velocity details restricted to only first few tens of meters below the Earth's surface. The velocities obtained towards the SE end of profile P1 (see Fig. 2b for location) are poorly constrained due to the one-sided ray propagation (no sources). Although, we still used this section of data to complement the illumination in the main survey area. We checked the quality of our FAT model by inspecting calculated first-arrival traveltimes with the picked first-arrivals for different shot gathers, assuring that majority of the traces
are not cycle-skipped. We used a smoothed version of the model for forward modelling, to avoid any strong heterogeneities produced by traveltime inversion and thus allowing a smooth energy propagation in depth. The smoothing was done by splitting the velocity model in two parts: top part with depth range between 0 - 250 m, and bottom part between 250 m – 1500 m. A Gaussian smoothing was applied with a shorter operator length on the top part to preserve the overall velocity variations. A larger operator length was used on the bottom part as the velocity model was not exhibiting detailed structures.






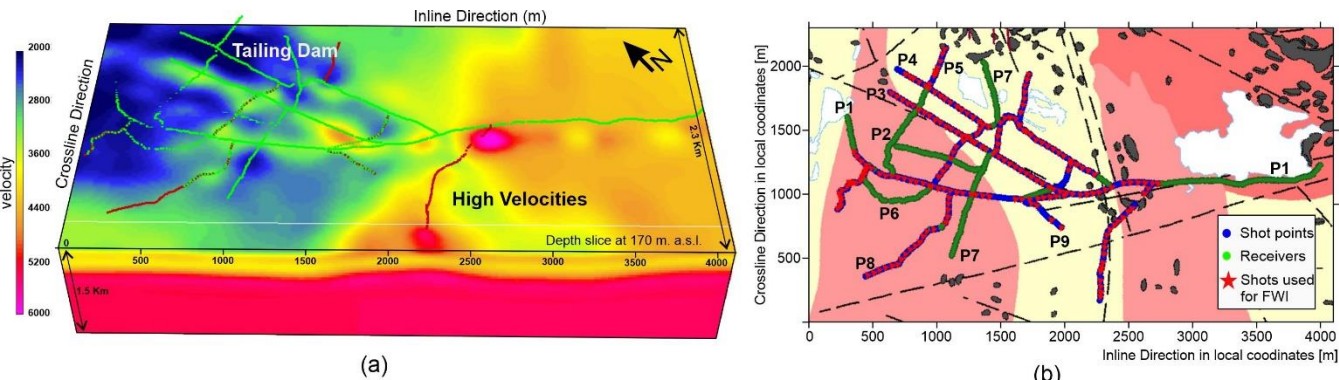

**Figure 2: (a) Starting velocity model for FWI built using FAT. The depth slice is shown at 170 m above sea level. The velocity model is highly heterogeneous at the shallow level; however, it is relatively homogenous at the bedrock level (few tens of meters below the surface). (b) Shots, receivers, and spatial extent used during velocity model building in FAT/FWI in the local coordinate system (blue rectangular box in Fig. 1). Green dots mark the receivers, overlapping blue dots show all the shots (>1000) and red stars show a subset of 216 shots used for FWI.**

### 3.1.2 Data preprocessing

Theoretically, FWI can start with the raw data. However, some signal processing is required to improve the SNR, especially at low frequencies, or to balance the frequency content. For acoustic FWI, it is also important to eliminate elastic effects, such as the surface waves. Our preprocessing is mainly focused on preserving the early arrival energy and improved signal coherency (compare Fig. 3a and 3b). A minimum-phase conversion was performed first, followed by surface-consistent trace balancing to average shot and receiver amplitudes due to variable near-surface conditions (Table 1). Then, a predictive deconvolution was applied to enhance the first arrivals, followed by FX-deconvolution for improved coherency and band-pass filtering (2-6-25-40 Hz) based on different frequency-band testing. A mute function was designed to remove the shear and surface waves. Finally, a trace normalisation was applied to provide equal representation to all offsets, effectively removing any viscoelastic responses. A comparison of raw data and data after pre-processing is shown in Figure 3. One can note that the first-arrivals are much better preserved with higher SNR, and improved coherency is achieved.

**Table 1.** Data preprocessing steps applied to raw data for FWI

| Data Preprocessing |
| --- |
| Read Data |
| Data conversion to minimum phase |
| Surface consistent amplitude balancing |
| Predictive deconvolution |
| FX-Deconvolution |
| Bandpass filter [2,6,25,40] |
| Muting (first-arrival based) |
| Trace Normalization |
| Write Data |


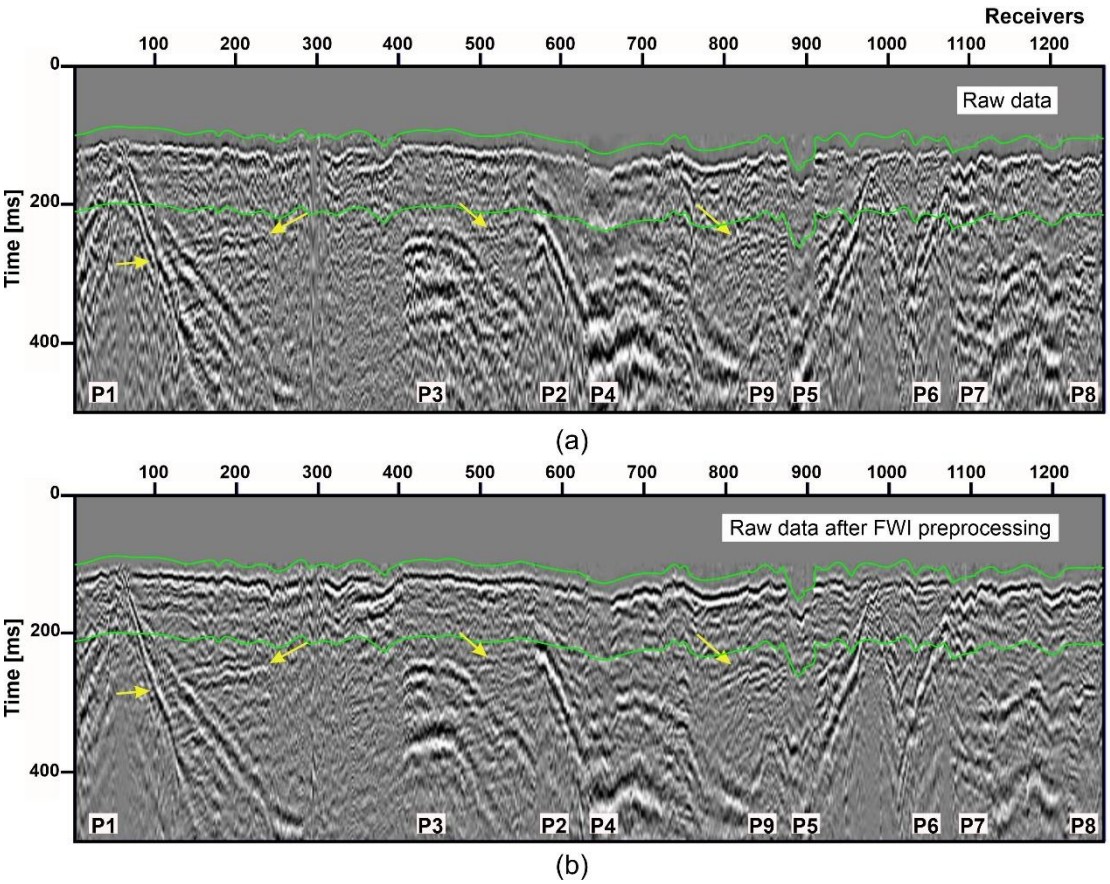

(a)

(b)

**Figure 3: A comparison of an exemplary (a) raw show gather, and (b) data after preprocessing (observed data) is shown after applying linear moveout (LMO) with a constant velocity of 5500 m/s and bulk shift of 100 ms. Yellow arrows mark the reflection from the mineralisation, and green lines show the data range used during FWI inversion. The location of different receiver lines marked with P's can be followed in Figure 2b. Note that after the preprocessing, first arrivals are more prominent with higher SNR and better coherency.**


### 3.1.3 Inversion parameters and strategy

Inversion parameters such as choice of optimization algorithm, type of gradient preconditioning and regularisation, data weighting and source wavelet estimation were thoroughly tested and fine-tuned accordingly. We inverted for the P-wave

velocity keeping a constant density during the inversion (2850 kg/m$^3$). Based on different frequency tests on the highly energetic early arrivals and their SNR response, we used a single frequency band (6-25 Hz) for the subsequent inversion. We used the SD optimization algorithm with approximate Hessian. L-BFGS optimisation has also been tested, but due to its higher rate of convergence, we encountered several artefacts yielding instability of the inversion. Also due to the presence of a lot of noise in the data, we decided to use SD optimization as its convergence rate is much slower and it is less likely to be trapped

in the local minima. Both the forward modelling and inversion were carried out on a uniform grid spacing of 10 m in each





direction - the same as for the resampled starting velocity model. We used a smoothed model topography obtained from the LIDAR survey in the area. We modelled a vertical single force source and vertical single force receivers (vertical geophones) without a free-surface. Data weighting and muting is implemented implicitly in the code.

**Shot selection, data weighting and source wavelet estimation**

As FWI is computationally very intensive, we need to find a good balance between processing power and memory bandwidth. In this study, we manually chose a subset of 216 good-quality shots out of more than 1000 shots available in the survey due to computational limitations (this was the amount fitting to 36 cluster nodes with 24 cores each, such that 4 cores were dedicated to one shot point). The criteria for the selection of shots were: good SNR, clear first-arrivals and uniform distribution within the survey area (red stars marked in Fig. 2b). Although we manually picked the preferred 216 shots, at a later stage we also performed the tests with random shot selections to quantify the effect of the shot grouping.

Since we aimed at using early-arrivals only to build our velocity model, we designed an external mute function to restrict the direct and shear waves. This is required to remove the part of data that contains the elastic effects, otherwise, the acoustic approximation will fail. Since trace normalisation is already applied to the data, we do not preserve amplitude variation with offset information anymore. To drive the model updates in the deeper section, we used data weighting of the misfit function equivalent to the absolute offset value of the trace.

The final part to start with the inversion was the estimation of the source wavelet following the linearized method by Pratt, 1999. During FWI of land data several factors like source coupling, receiver coupling, local ground condition etc., significantly affect the characteristic of the source wavelet, making it difficult to derive the correct source signature for the modelling. Here we essentially tested two strategies: (i) a single average source wavelet estimated using all the shots (216) (see estimated source wavelet in the lower-right side of Fig. 7), and (ii) individual source wavelets for each shot point (Fig. 8). In both cases, source wavelets resembled the minimum-phase equivalent of the Vibroseis sweep signature. Before being used in FWI, source wavelets were bandpass filtered and scaled to match the amplitude of the observed data. However, after several tests with individually estimated source wavelets, we concluded that scaling and handling each wavelet separately to match the amplitudes of the observed data was difficult, and was producing artefacts in the velocity model. Therefore, we decided to use the average source wavelet which was also additionally scaled to match the observed data. We also tested the scenario where the average source wavelet is re-estimated after every 10 FWI iterations. However, there were no significant changes in the estimated source wavelet signature from one cycle to another. In the end, we observed that this exercise did not contribute to a significant change in the final velocity model as well comparing to the approach where the wavelet is kept the same for the whole inversion. Therefore, we decided to follow the latter approach. All the results presented afterwards in this article are produced using this approach.





### 3.1.4 FWI results

Due to computational limitations, we were unable to process all the 1000 shots from the survey at the same time. The baseline dataset comprised of manually selected best-quality (and relatively uniformly distributed) 216 shots. In the next stage, three different subsets of randomly selected 216 shots with a uniform distribution within the survey area were used in FWI (Fig. 4). In this section, we present the results obtained from both approaches. We started with the general approach of FWI. As FWI is a local optimization technique, we used the velocity model produced from FAT as starting model (Fig. 5a). We used a single source wavelet, the constant density of 2850 kg/m³, SD optimization algorithm and smoothed Hessian to build a P-wave velocity model. We checked the quality of the velocity models based on data fitting, wavelet estimation, drop in the cost function, comparison with other results obtained from direct measurement in boreholes and visualisation.

**Subset of manually selected shots**

Smoothing is applied in each iteration to the gradient before it is scaled to obtain model perturbations which are then added to the current velocity model. An approximately ~18% reduction in the cost function is observed in the first 40 iterations after which the drop was still monotonously decreasing but negligible (light blue line, Fig. 4). From Figure 5, we can infer that the velocity details in FAT (Fig. 5a) are restricted to the first few tens of meters from the surface, otherwise, it is almost a 1D velocity model in depth. On the other hand, the velocity model from FWI (Fig. 5b) is characterised by velocity details to ~1000 m in depth.

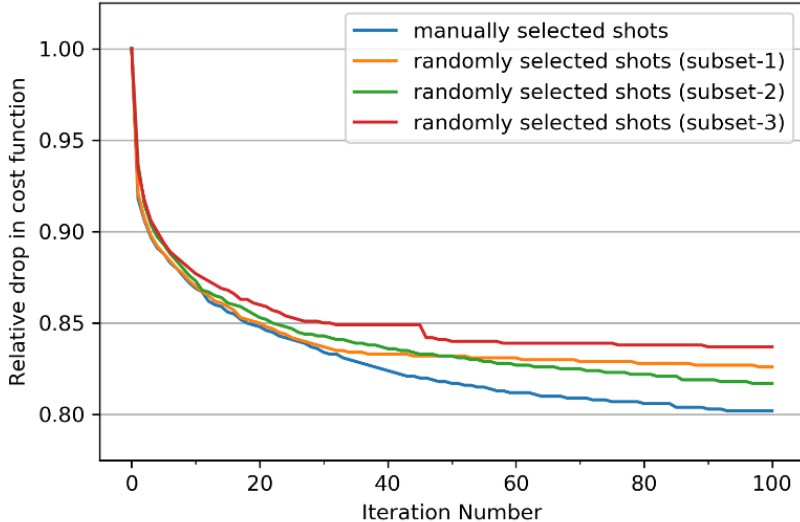

**Figure 4: A plot showing the relative drop in cost-function for the inversion using different shot subsets. All the plots show that the inversion strategy is stable and effective, and does not depend significantly on the selection of shots as long as the uniform areal distribution of shots is followed across the survey area.**





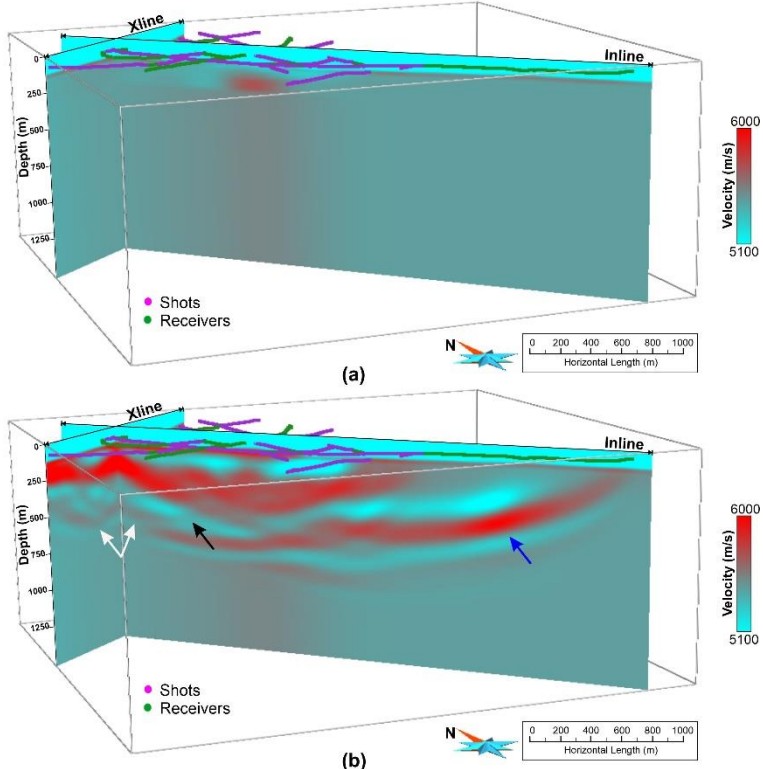

**Figure 5: Comparison of velocity model resulted from (a) FAT, and (b) FWI using a dataset comprising a manual selection of 216 shots. Note that the velocity details with depth for the FAT velocity model is restricted in the near-surface region while FWI derived**
**velocity model has much greater details at depth. White arrows indicate a dipping high-velocity layer in the SE direction which appears to follow a curved geometry in the SW direction, black arrow shows the possible presence of a cross-cutting fault and blue arrow shows artefact introduced in the velocity model due to only one-way energy propagation as there are only receivers in the SE part of the survey. Several other features in terms of high and low velocities can also be inferred in the near-surface region.**

**Subset of randomly selected shots**

Further, in order to validate our inversion strategy, we randomly selected three different subsets each containing 216 shots with uniform distribution in the survey area as previously done. The idea here was to see the effect of a random selection of shots compared to the manual selection of best quality shots on the inversion strategy. A large difference in the velocity model produced from both approaches would suggest that more emphasis on data selection has to be given, and a more detailed
investigation on inversion strategy is required. Here, we kept the same configuration as followed in the previous section to produce our preferred model. The velocity model is produced for all the three subsets with a similar drop in cost function and convergence (see comparison in Fig. 4). The velocity model obtained from a subset of randomly selected shots (subset-2, Fig. 4) is shown in Fig. 6a (compare with Fig. 5b). Both the velocity models show similar characteristics in terms of different features that can be observed (compare marked arrows, Fig. 5b and 6a). Figure 6b and 6c shows velocity perturbation for two
different subsets, i.e. subset-1 and subset-2 with respect to the model produced from a manual selection of shots (Fig. 5b). We observed an average velocity difference of around ±50 m/s (Fig. 6b and 6c, also for subset-3) in the area which is well





illuminated while a large difference is observed where sampling is poor or velocity model is less-constrained i.e. on the edges

of the survey. A histogram plot shown in Figures 6d and 6e (for models shown in Fig. 6b and 6c, respectively) is also produced

to understand the velocity perturbation quantitatively. One can note that the majority of the points are clustered within the

displayed range (±50 m/s), whereas the total number of points outside this range constitutes less than ~6% of the total points.

These comparisons indicated that our inversion strategy is effective & stable, and does not rely substantially on the shot

selection as long as their uniform areal distribution is followed.

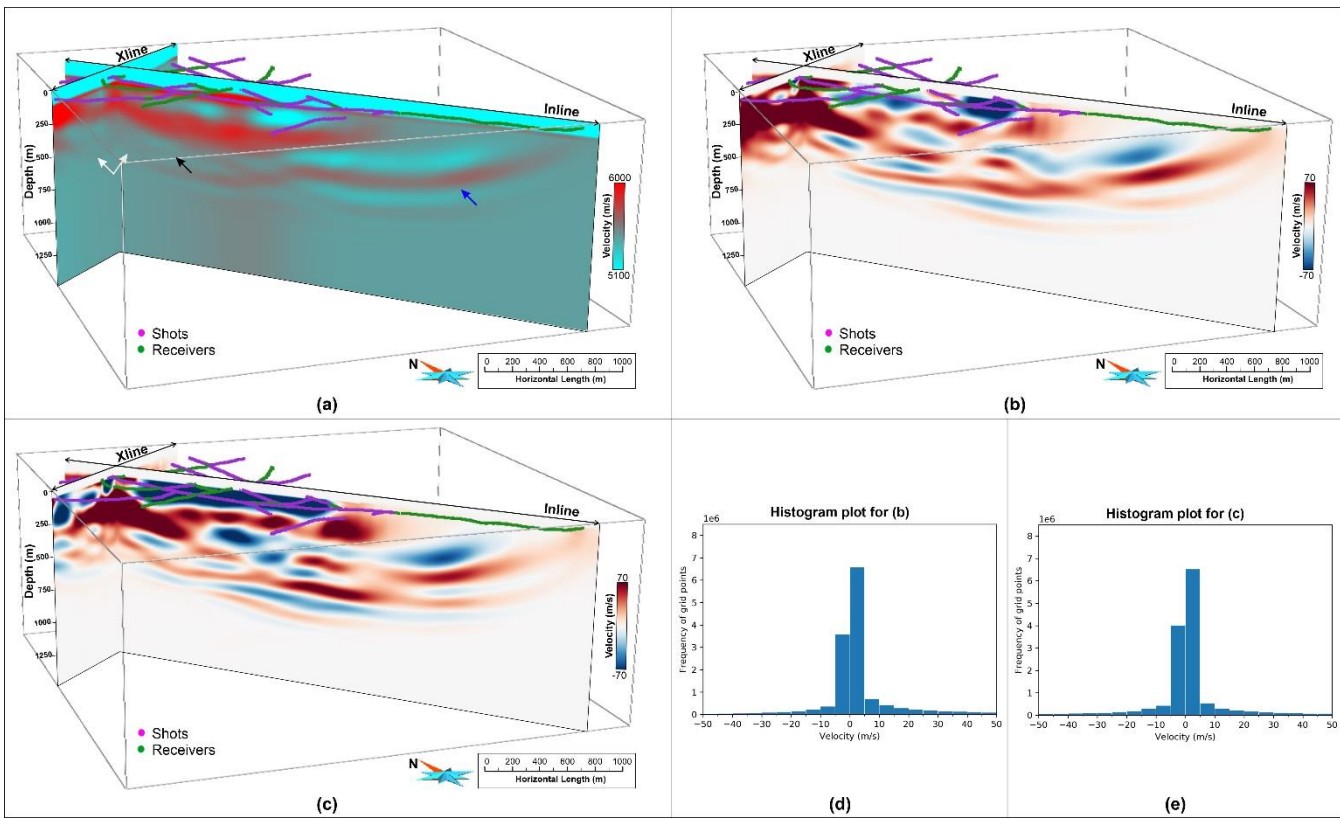

**Figure 6: (a) Velocity model produced from random selection of shots (subset-2), compare with Figure 5b, (b) velocity perturbation model produced using velocity model derived from subset-1 and from manual selection of shots, i.e. velocity model shown in Figure 5b, (c) same as (b) but for subset-2, (d) and (e) histogram plot for (b) and (c). The plot shows the efficacy of the inversion strategy which effectively produces a similar velocity model independent of shot selection within acceptable limits of velocity perturbation.**

### 3.1.5 FWI results assessment

In order to check the accuracy of the velocity model, we assess the data fitting between observed and synthetic gathers, wavelet

estimation and cost-function drop. We also confronted our velocity model with *a priori* information and other available results

in the survey area. Here, we are presenting the result assessment for our preferred velocity model only (Fig. 5b).





**Real versus synthetic data comparison**

Data fitting of common-shot & common-offset (CO) sections between observed data and synthetics produced from the FWI
velocity model is shown in Figure 7. A CO section is produced by selecting different source-receiver pairs within a fixed offset
distance. In comparison to a common-shot gather where only a single shot can be evaluated at a time, CO sections enable to
display information from all the inverted shots at once. This way all the shots can be evaluated simultaneously for different
offsets. We computed the CO section with a bin width of 50 m and bin-centred sections produced every 250 m. Final CO

sections are produced for the data range used during the FWI after applying linear-moveout velocity of 5500 m/s and a bulk
shift of 100 ms. A common-shot gather comparison between observed data and synthetics is shown in Figure 7a. Based on
different shot gather comparisons, we noted that the overall fitness of the data is good with some localised areas susceptible to
cycle-skipping in short to mid-offset ranges. For far offset traces, the velocity model was only able to find a partial fit in some
cases, such as shown by the yellow arrow in Fig 7a. It is most likely inherited from the starting model where it was locally

unable to provide a kinematically good fit to first arrivals. Three different CO sections for bin-centred at 250 m, 1000 m and
1500 m are shown in Figures 7b, 7c and 7d. Different CO sections at various ranges show overall good data fit for at least the
first-cycle of the waveforms for a majority of shot points. Local cycle-skipped positions are marked by yellow arrows in Figure
7b, 7c and 7d for different shot points. It is likely to be inherited by the fact that statics correction had not been applied during
the data preprocessing prior to FWI.


**A posteriori wavelet estimation**

Another diagnostic of the robustness of the FWI-derived velocity model is the quality of the source estimation in the final
model. In Figure 8, we are showing wavelet estimation for all 216 shots for the initial model obtained from FAT (Fig. 5a) and
FWI velocity model (Fig. 5b). It can be inferred that the estimated wavelets from the FWI velocity model produce more

coherent signatures with better amplitude responses. Shot locations for which low amplitude wavelets are estimated (missing
gaps) belong to the area where the tailing dam is located (see Fig. 2 for location).

**Cost function drop and RMSE maps**

Another way of assessing the quality of the velocity model is to check the cost function convergence with each iteration. From

Figure 4, for all the cases, a drop-in cost-function is observed until the 40$^{th}$ iteration by large, after which the convergence was
minimal. To quantify the contribution of individual shot gathers to cost-function, we calculated root-mean-squared error
(RMSE) on a trace-by-trace basis. In Figure 9, we present the RMSE plots for two shot gathers. We show the evolution of the
data fit for the starting model (0$^{th}$ iteration), after the 10$^{th}$ iteration (up to which the most significant drop in the cost function
is observed, Fig. 4) and at the 50$^{th}$ iteration. An initial observation of the RMSE maps shows that the drop in the cost-function

is mainly driven by the traces present in the near-to-intermediate offset ranges (compare traces marked by blue arrows for
different iterations in Fig. 9). The traces present in the intermediate-to-far offset range has comparatively less contribution in



the reduction of cost-function. It might be due to the fact that the starting model was not able to produce the kinematic fit to first arrivals at far-offset ranges as well as because they are least-constrained due to their presence at the edge of the survey.

**Figure 7: (a) Data fitting comparison between observed data (black and white) and synthetic data (red and blue) produced from FWI velocity model (Fig. 5b) for a common-shot gather. (b), (c) and (d) shows common-offset (CO) sections for bin-centered at 500 m, 1000 m and 1500 m. Source wavelet used during FWI is shown on the bottom right. The overall fitness of the data is acceptable, unless for mid-to-far offset ranges where data fitting is either partly fit or is prone to local-cycle skipping (yellow arrows). CO sections shown here are for the data range used during FWI after applying a linear-moveout correction with velocity 5500 m/s and a bulk shift of 100 ms. Note that the data fitting between observed and synthetics for the first-cycle of the waveform is good while for the second-cycle, there are places where the waveforms are partially overlapping or local-cycle skipped.**





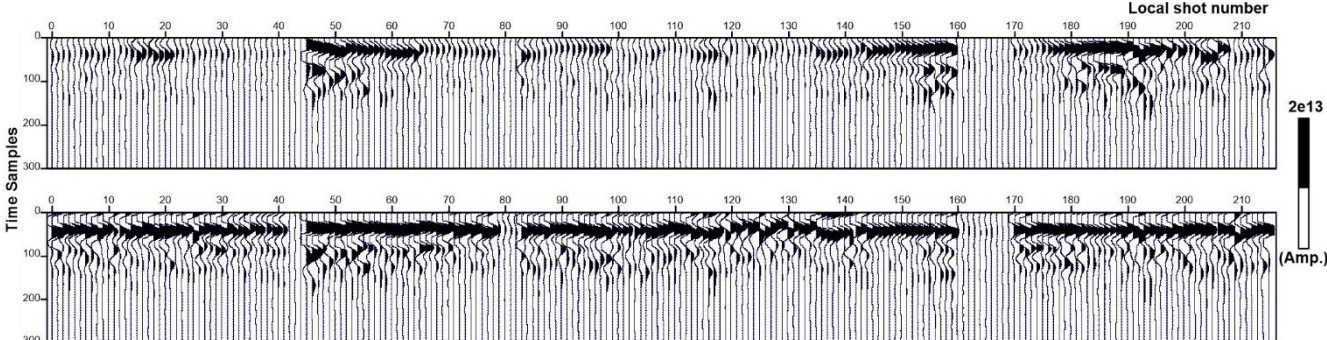

**Figure 8: Source wavelet estimation for each shot location used in FWI for (a) starting model from FAT, and (b) FWI velocity model. Note that a better amplitude response and coherency is obtained from the FWI velocity model.**

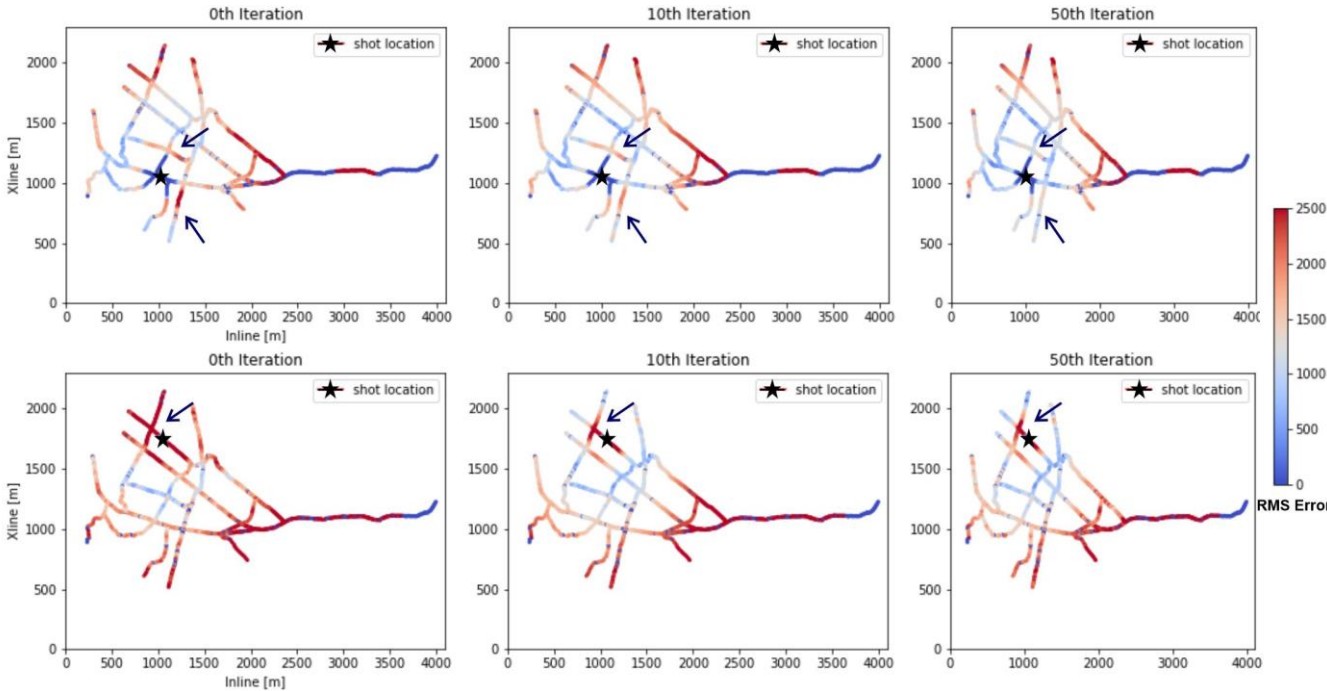

**Figure 9: RMSE maps for two different shot locations showing the drop-in cost-function at different iterations. The drop in cost-function is mainly driven by near-to-intermediate offsets traces, while far offset traces has comparatively less reduction. Zero value corresponds with the traces which were omitted during the FWI.**

## 3.2 Reverse time migration

The complete solution to seismic imaging consists of two main parts: first, building a long-wavelength velocity model, and second, obtaining reflectivity structures using seismic migration. In this section, we are discussing our approach to imaging Ludvika 3D seismic data using RTM. The overall aim of RTM was to validate the FWI velocity model and clearly delineate





the dipping reflector along with other plausible geological features in the survey area. We compared RTM stacks obtained for three different velocity models: a constant velocity model of 5600 m/s, the smoothed FAT model and the FWI model.

### 3.2.1 Data preprocessing

The data used for RTM were processed in a similar way as discussed in Hloušek et al. (2021). The processing was mainly aiming at the suppression of surface waves and improvement of reflected signals associated with the mineralisation (Table 2). Refraction static corrections were calculated and applied to the data in two different ways. In the case of migration using the constant velocity model, a generalized refraction traveltime inversion approach was used (GLI3D, Hampson and Russell, 1984). In the case of RTM with the FAT and FWI velocity models, a tomostatics approach was used with the same velocity

model as used as starting model for FWI (Fig. 5a), however, only the residual part of the statics was actually applied to the data.

Table 2: Data processing applied to the Ludvika 3D dataset for depth imaging

| Processing parameters | |
|---|---|
| Amplitude normalisation | Surface-consistent for shots and receivers |
| Minimum-phase conversion | Based on matching filter using theoretical sweep |
| Refraction statics | GLI3D or tomostatics |
| AGC | 200 ms window length |
| Spiking deconvolution | 80 ms operator length, single trace |
| Bandpass filter | 15-35-145-165 Hz |
| Surface-wave attenuation | Wavelet-transform based (v ≤ 2700 m/s) |
| FX-deconvolution | Yes |
| Amplitude scaling | Whole-trace RMS amplitude balancing |
| Top mute | 30 ms below the picked first-arrivals |

### 3.2.2 Implementation and computational aspects

We used a RTM algorithm implemented in Shearwater Reveal software to run 3D RTM using our 3D dataset consisting of 1044 shots. We used a minimum-phase Ricker wavelet with a peak frequency of 70 Hz as a source wavelet based on an average medium velocity of 6000 m/s. An isotropic wave propagation was modelled with $4^{th}$ order in space and $2^{nd}$ order in time using finite-difference operators. A convolutional perfectly matched layer (CPML) boundary condition was used with 12 grid points in thickness and 8 grid points for padding at the boundaries. A standard zero-lag cross-correlation was used as the imaging

condition. The inline and crossline aperture was fixed to 1 km and 1.8 km, respectively. A 10 % aperture taper was used to suppress the migration noise on the edges. The time step and grid size were automatically adapted to the velocity model (see Table 3). However, migrated shot gathers were produced with a grid spacing of 10 m, the same as the input velocity model. The final RTM stack was produced by accumulative stacking of all migrated shot gathers. Only a low-cut filter was applied to





the stack to remove the near-surface low-frequency noise typical for many RTM implementations. RTM was run in parallel
mode at our local cluster. It took ca. 8.5 hrs to produce the final result for the constant velocity model using 7 nodes of Intel(R)
Xeon(R) processor each containing 24 cores. For the FAT and FWI case, it took ca. 20.8 and 28.5 hrs respectively.

Table 3: Time step and grid size information for RTM computation

| Parameters | Const. vel. model | FAT model | FWI model |
|---|---|---|---|
| Time step (ms) | 0.8 | 0.56 | 0.6 |
| Grid size (m) | 9.0 | 6.42 | 7.0 |

### 3.2.3 RTM results

RTM with a constant velocity model was able to highlight a dipping reflector in the SE direction, which follows a curved
nature in the SW direction with a hint of cross-cutting fault dipping in opposite direction (see yellow arrows in Fig. 10a). On
the other hand, RTM with the FAT velocity model further improves the reflectivity of mineralisation and clearly highlights
the termination of the dipping reflector by a cross-cutting fault (Fig. 10b). The depth image otherwise is very noisy in the near-
surface area. RTM with FWI velocity model produces a depth image with much better focussing of the dipping reflector and
a clear representation of the cross-cutting fault which appears much deeper in depth towards the West and to the surface in the
East. (Fig. 10c). The depth image with the FWI velocity model also highlights other reflectors, normal and cross-cutting faults
in the near-surface section which is significantly more noisy for the previous two results. The image also has less migration
noise which comprehends the fact that a detailed velocity model can be a great asset in producing accurate subsurface images.

To further understand the depth extent and geometrical nature of the dipping reflector associated with mineralisation, the 3D
cube was investigated in more detail. Figure 11 shows successive slices in depth, crossline and inline direction. In the depth
slices (Figure 11a-11e, left panels), the reflectivity related to the mineralisation can be tracked comfortably down to the depth
of 1000 m. Similarly, depth images along the crossline direction (middle panels, from NW to SE direction) show the curved
nature of the mineralisation clearly in the SW direction which was earlier believed to be flat. After almost crossing the middle
of the survey area from SW to NE, a second prominent reflector below the mineralisation appears to be in place until the end
of the acquisition line in the NE direction (middle panels, Figures 11d-11e). The inline sections (right panels) confirm the
progression of the mineralisation at depth until it breaks off at a major cross-cutting fault (Fig. 11c). The extent of the
mineralisation can be easily followed from the NW to SE direction.






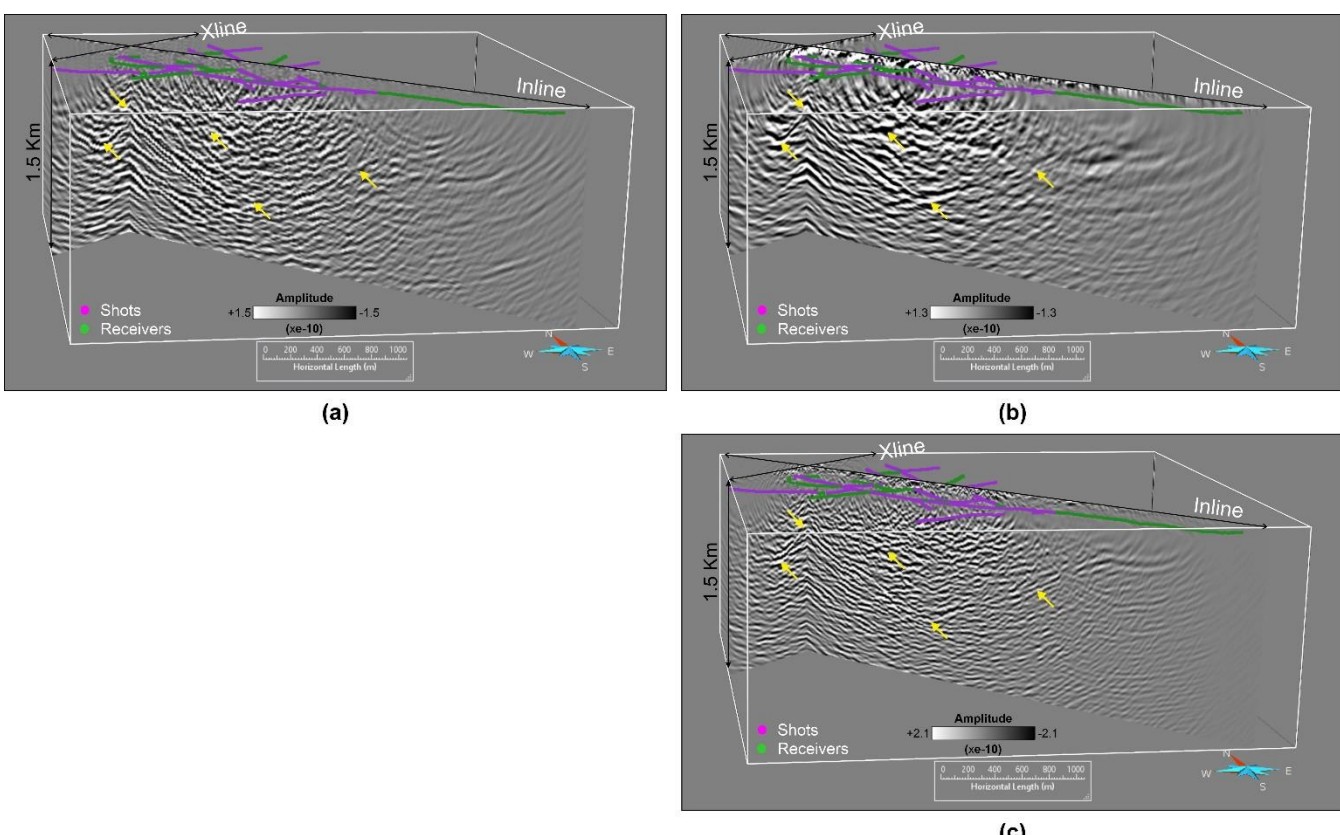

**Figure 10: Comparison of the depth image cross-section produced from RTM using, (a) a constant velocity model of 5600 m/s, (b) FAT model, and (c) FWI velocity model. The depth image is referenced to 250 m above sea level (same as velocity model). Yellow arrows highlight different features observed in the depth images. One can interpret that the depth image produced using the FWI velocity model is much more focussed, is less noisy and significantly improves the imaging in the near-surface section as compared to the other two images.**




**Figure 11: Depth (left panels, a-e), crossline (middle panels, a-e) and inline (right panels, a-e) sections through the final RTM stack using the FWI velocity model. The red dashed line shows the extent of the FWI velocity model used in RTM. Blue lines show inlines and crosslines at different positions. The yellow arrow shows some prominent reflectors observed in the RTM stack cube.**

## 4 Interpretation and discussion

The overall aim of the 3D survey was to better understand the geometry of the deposits as well as to better constrain structural features of the host rock and associated discontinuities. We produced a high-resolution P-wave velocity model using FWI. A


cross-sectional view of the obtained velocity model is shown in Figure 12. To validate the reliability of our velocity model,
we compared our results with the geological model of the known mineralisation mainly based on the drillholes. A good
correlation was found between the dipping high-velocity layer and the known mineralisation shown in Figures 12c and 12d.
We can interpret a dipping high velocity layer in the SE direction (blue arrow, Figures 12a and 12c) resembling the shape of
the known mineralisation. A previously modelled ore lens appears to follow a curved geometry in the SW direction, whereas

the velocity model suggests an up-dip continuation of the high-velocity layer in the NE direction. A high velocity filled zone
in a basin form is visible in the shallower section along with the hints of several geologically plausible fault-like structures
(black arrows, Fig. 12c). An artefact in the form of a layer filled with high velocities is also indicated by the red arrow in
Figure 12c due to the fact that there are only receivers on this end of the survey and the energy propagation was only one-way.
The above examples suggest that the detailed velocity model produced using FWI can serve as an independent asset for

interpretation. Such details cannot be inferred from the smooth FAT model.

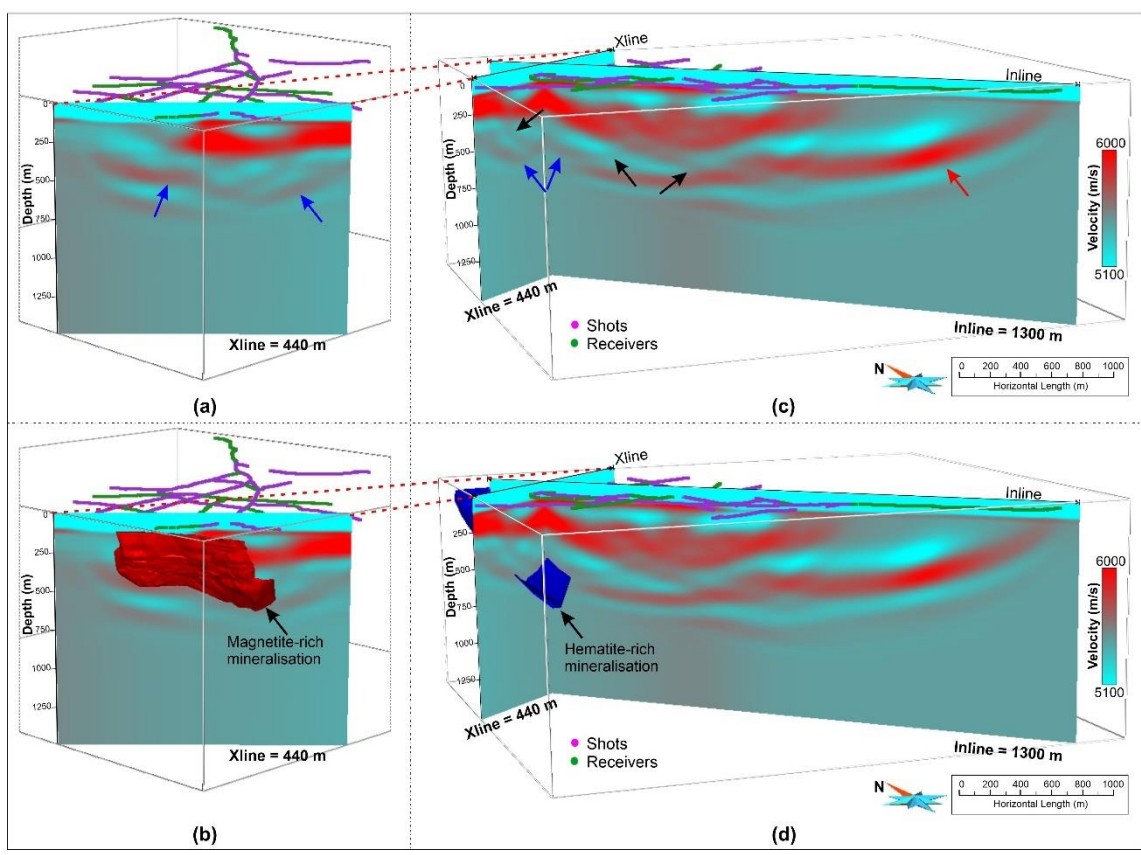

**Figure 12: (a, b) Crossline view of the FWI derived velocity model, (c, d) Cross-sectional view of the FWI derived velocity model. Projection of the modelled mineralisation is shown in (b) and (d). Blue arrows mark high-velocity layers interpreted to be associated**
**with mineralisation, black arrows show different geologically plausible fault structures and the red arrow shows artefact from FWI due to the one-sided illumination.**



Another important aspect of our study was to ultimately test whether a high-resolution velocity model built using FWI yields a better and more accurate depth image than the one obtained using a smooth FAT model. Figure 13 shows a comparison of

the depth images produced from RTM using the velocity model derived from FAT (Fig. 13a, 13b and 13c) and FWI (Fig. 13d, 13e and 13f). RTM using the FAT velocity model was able to map the reflector dipping in the SE direction and highlight its curved 3D geometry in the SW direction, and suggesting that it continues up-dip in the NE direction (red arrows, Fig. 13a and 13b). When compared with the modelled mineralisation, the associated reflector shows a good agreement in terms of both position and shape (blue and pink surfaces, Fig. 13c). Another package of reflections roughly ~250 m below the main

mineralisation was also delineated (blue arrows, Fig. 13a and 13b). A major cross-cutting fault appears to be restricting the downward continuation of the mineralisation with depth (black arrows, Fig. 13b). There are several indications of fault-like structures in the near-surface region but they are otherwise very noisy to clearly follow their continuation (yellow arrows, Fig. 13a and 13b). All these events can be followed in the depth images produced using FWI derived velocity model (compare Fig. 13a, 13b and 13c with 13d, 13e and 13f). The first impression from this comparison suggests that a more focussed image is

obtained using the FWI velocity model. Reflector associated with the mineralisation has now better focussing and fitting in the down-dip direction (compare Fig. 13c and 13f), also its up-dip continuation in the NE direction is more clearly delineated (compare red circles marked in Fig. 13a and 13d). The intersection of cross-cutting fault with mineralisation is more distinctly established and its extent both in up-dip and down-dip direction is more clearly delineated (compare Fig. 13c and 13f). Also, the presence of several faults in the near-surface can now be followed more clearly (compare Fig. 13b and 13e). Overall, the

depth image based on the FWI velocity model is less noisy with higher accuracy, which clearly indicates the superiority of using a high-resolution velocity model in the wavefield extrapolation depth migration such as RTM.

We also compared noticeable features present in the FWI velocity in terms of high and low velocities with its corresponding RTM depth image. Figure 14 shows such a comparison for two different inline positions (compare 14b and 14c with 14e and

14f) while keeping the same crossline position (compare Figures 14a with 14d). Different events marked by black arrows in the FWI velocity model corresponds to fault structures imaged via RTM. This depicts the accuracy of our built model and further confirms the inference that FWI derived velocity model can also be used as an independent interpretation tool.

The 3D dataset used in the current study has also been the subject of a conventional processing (time-domain) workflow to

provide a first-hand geological interpretation of the study area (Malehmir et al., 2021) as well as of an advanced focusing Kirchhoff PreSDM for depth imaging (Hloušek et al., 2021).  A comprehensive comparison of our results with other studies previously done in the area (including depth imaging along the P1 profile by Bräunig et al. (2020) or Ding and Malehmir, (2020)) is beyond the scope of this paper and will be subject of a separate follow-up paper.





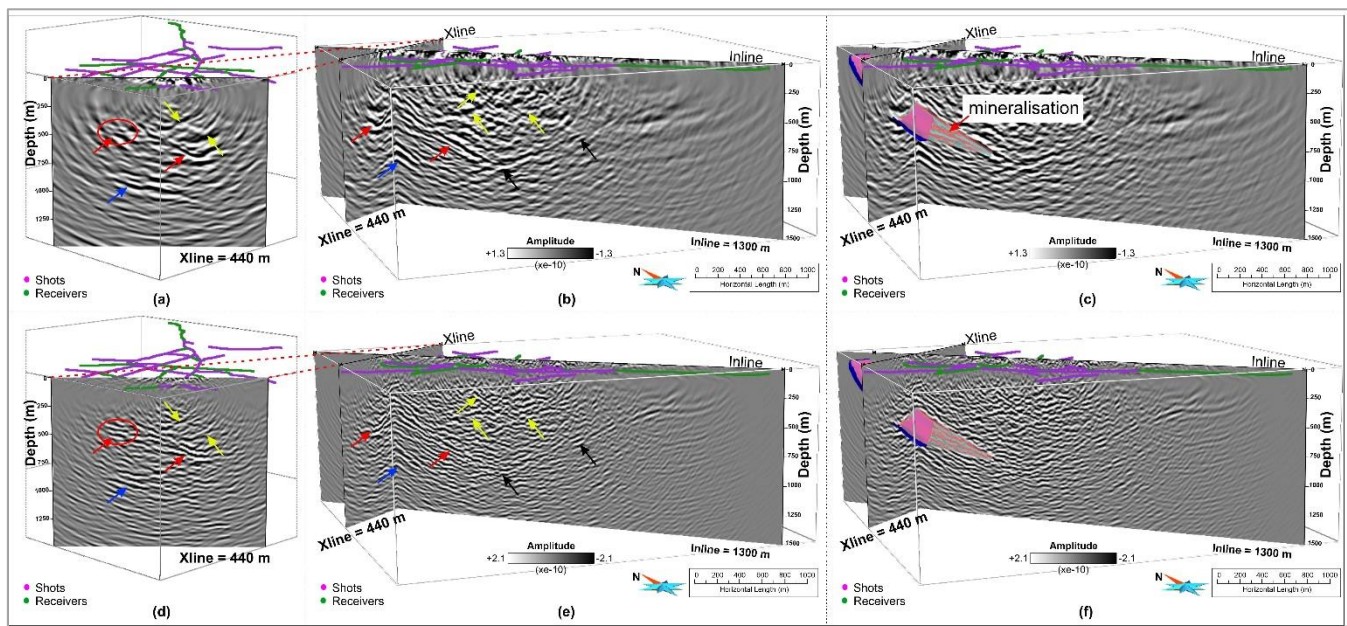


**Figure 13: (a) and (b) shows a crossline and cross-sectional view of depth image produced from RTM using the FAT velocity model. (c) is same as (b) with the projection of know mineralisation. (d), (e) and (f) is same as (a), (b) and (c) but for FWI velocity model. Different arrows show different events observed in the depth images. Blue and pink surfaces are the known mineralisation surfaces produced mainly based on drilling in the area. The depth image derived from the FWI velocity model is less noisy, more focussed**

**and with higher accuracy as compared to the FAT velocity model.**

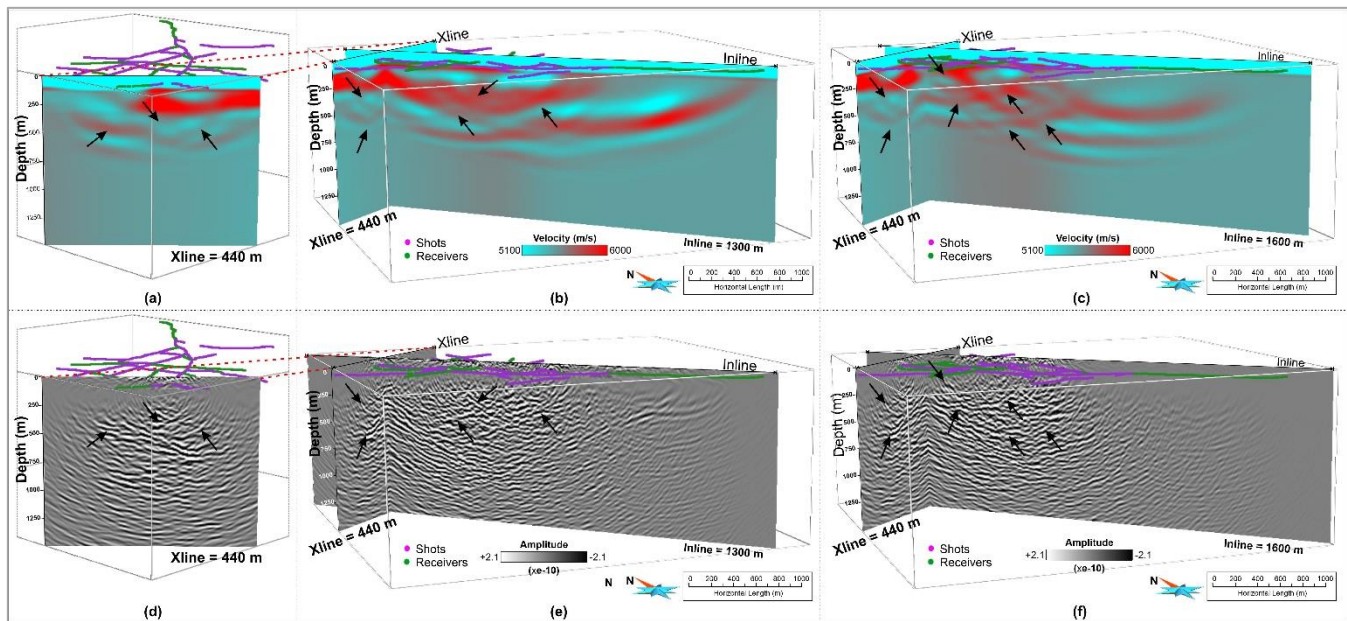

**Figure 14: (a) shows crossline, (b) and (c) and cross-sectional view of FWI velocity model for two different inline position. (d), (e) and (f) is same as (a), (b) and (c) but for RTM depth image. Black arrows shows noticeable events present for both the results.**




Our case study provides a foremost initial understanding of the advantages and shortcomings of applying joint FWI-RTM imaging workflow in a hardrock environment and forms the basis for future works. On the acquisition side, more regular survey designs with longer offsets and better azimuthal coverage would make FWI more feasible, but bear the risk of introducing acquisition footprints into the resulting models and images. The incorporation of reflection modes in conjunction
with diving/refracted rays will reduce the dependency on the longer offsets and produce high-fidelity velocity models. Mono-parameter to multi-parameter inversion, choice of the norm in the misfit function (e.g., L2 vs optimal mass transport), the role of the density and acoustic to elastic wave-equation based FWI should also be investigated. Higher velocities in the near surface and steep velocity contrasts in hardrock environment easily produce numerical dispersion, therefore finite-element or spectral-element methods should be tested in place of current FD method. On the imaging side, different imaging conditions
in RTM could be explored, together with the inversion formulation of the migration (least-square RTM) for more appropriate amplitude handling.

**5 Conclusions**

We have demonstrated a joint imaging workflow consisting of velocity model building step by FWI and depth imaging by RTM using a fixed-geometry sparse 3D seismic data acquired over Ludvika mines in central Sweden. We have developed a
data pre-processing workflow and a FWI strategy for building a high-resolution velocity model in hardrock environment. We obtained a high-fidelity 3D velocity model cube with greater details to ca. 1000 m depth as compared with the FAT model where the details are limited to just a few tens of meters. We also applied and thoroughly tested RTM for subsequent depth imaging. The FWI derived velocity model produced the most focussed and accurate depth image compared to constant velocity and FAT velocity models. The known mineralisation was clearly delineated down to ca. 1000 m depth with details on its 3D
shape. A major cross-cutting fault was mapped, which appears to be restricting the extension of the mineralisation at depth. Different faults were also delineated in the survey area, which were earlier dismal or unknown with such accuracy. We advocate that the combination of FWI and RTM is highly beneficial for subsurface imaging in the hardrock environment. Although both methods are computationally more expensive with respect to standard practice i.e., time-domain or ray-based imaging, it is worth investing in them, particularly where the detailed subsurface image is required, e.g., for resource
identification and improved depth targeting for drilling.

**Data availability**

Data associated with this research are available per request to project coordinator Alireza Malehmir (alireza.malehmir@geo.uu.se), Department of Earth Sciences, Uppsala University, 75236, Uppsala, Sweden.



**Competing interests**

The authors declare that they have no conflict of interest.

**Special issue statement**

This article is part of the special issue "State of the art in mineral exploration". It is a result of the EGU General Assembly
2020, 3–8 May 2020.

**Acknowledgement**

First and foremost, we thank Magdalena Markovic from Department of Earth Sciences, Uppsala university in survey design
planning and preparation of acquired data. We also thank R. Brossier (ISTerre) and L. Metivier (ISTerre/LJK) for providing
us with the TOYXDAC_TIME FWI code developed in the frame of the SEISCOPE Consortium (available at
https://seiscope2.osug.fr). Globe Claritas™ under the academic license from Petrosys Ltd. and Seismic Unix was used for the
data processing. GeoTomo Inc. TomoPlus software was used for the traveltime tomography & refraction statics calculation.
We thank Shearwater Geoservices for granting us the academic license of Reveal software to run RTM. GOCAD® was used
for 3D visualization and sponsored by Emerson Paradigm. We thank all the participants from Uppsala University, Geopartner,
TUBAF, TU Delft and Nordic Iron Ore who had participated in the fieldwork especially the young professionals.
Computational resources were provided in part by the PLGRID HPC infrastructure.

**Funding**

Smart Exploration has received funding from the European Union's Horizon 2020 research and innovation programme under
grant agreement no. 775971. The Vibroseis source of Technische Universität Bergakademie Freiberg has been funded by the
Deutsche Forschungsgemeinschaft (DFG) under grant no. INST 267/127-1 FUGG.

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
