# Peer review of "3D high-resolution seismic imaging of the iron-oxide deposits in Ludvika (Sweden) using full-waveform inversion and reverse-time migration"

_Solid Earth, 2021_

## Author Comment (AC2)

**Reviewer 2**

First of all, we would like to thank the reviewer for his comments and suggestions. Below we list our detailed answers. Please note that the mentioned line numbers refer to the original submission.

**GENERAL COMMENTS**

You should complement the introduction with references to studies from other research groups and not only self-citations, particularly in these sentences outlining general aspects. For example, you could illustrate the examples of use of FWI in hydrocarbon exploration outlined in L62 with apropriate references to different studies (and maybe add other relevant challenges addressed, such as sub-salt or sub-basalt imaging).

**Reply:** This point was a bit unclear to us. Regarding the 'general' introduction, it is hard to avoid self-citation aspect, as the hardrock seismic community is relatively small and many members of this community are in fact contributing to this manuscript. Best to our knowledge, this is probably the first application of a joint imaging workflow consisting of FWI and RTM in the hardrock environment. Similarly, we cannot avoid same issue when talking about the depth imaging approach (L49), but we tried to supplement it with other references, for example, Schmelzbach et al. 2008.

Regarding the industry FWI examples: we talk first about "solving complex imaging challenges, e.g. seeing through gas clouds and resolving shallow velocity heterogeneities" (L61), which is mostly relevant to marine environment (similarly with the sub-salt, sub-basalt imaging) and that's why we do not cite any references. Then we turn the reader attention to land data, citing some relevant papers:

"Nevertheless, a few successful case studies have been reported for 2D and 3D land datasets using acoustic/viscoacoustic FWI (Ravaut et al., 2004; Malinowski et al., 2011; Baeten et al., 2013; Adamczyk et al., 2014; Stopin et al., 2014; Cheng et al., 2017)".

I am not a fan of describing the paper organisation in the introduction (last paragraph of the introduction), as it ends up being very redundant (e.g. "Finally, we conclude our case study in the 'Conclusions' section" is quite obvious). The one but the last paragraph looks much better as a wrap up of the introduction section, so I'd suggest removing the last paragraph entirely.

**Reply:** Thank you for this suggestion. We have removed the last paragraph from the introduction part.

L197 - It would be useful to show a map of the raypath coverage, to give a hint of the areas with higher or lower uncertainty. In particular, areas to the SE of the study area do not seem to be sampled by any raypaths; the model in these areas is the initial or an extrapolation from the nearby locations?

**Reply:** We have revised Figure 2 where we are now showcasing different depth slices from the tomographic velocity model, masked according to the ray coverage.

**SPECIFIC COMMENTS**

L34 - "the need for it", explain the "it", use "this technique" or so.

L36 - bringS L93 - What does "endowment" mean in this sentence?

L102 - The references should be ordered either by date of publication (preferrable) or alphabetically, but you have both systems. Please check throughout the manuscript.

L106 - What other methods are included within that "etc"? Please explain.

L2014 - "Theoretically, FWI can start with the raw data" - this sentence is too vague, please explain. L261 - Does this refer to issues generally observed in land FWI studies or to specific issues observed in your study? If the former, add references; if the latter, please clarify.

L444 - You could remove "One can interpret that"

**Reply:** Thank you very much for highlighting these. All mentioned points are incorporated in the revised manuscript.

**FIGURES**

Figure 10 - Maybe this is a visual effect, but the depth image corresponding to the FWI image (10c) looks like having a higher frequency content than the other two. Is this the case?

**Reply:** Yes, this is correctly pointed out that the FWI image is having higher frequency content than other two. As RTM is computationally very intense, we decided to run it with the optimal time-step and grid-size based on velocity model used [Line 412, Table 3].

Figure 12 - Please specify in the caption where do the mineralization bodies interpretations come from (otherwise it looks like you are interpreting these bodies from the FWI model).

**Reply:** This is also incorporated in the revised manuscript.

**REFERENCES**

Many of the references are incomplete (e.g. lacking th journal name), please check and amend them.

**Reply:** Thank you very much for the fine observation. All such discrepancies had been removed in the revised manuscript.

---

## Author Response (AR1)

**Reviewer 1**

First of all, we would like to thank the reviewer for his comments and suggestions. Below we list our detailed answers and changes in manuscript. Lines numbers mentioned here are according to new revised manuscript or otherwise stated. Changes in the text in the revised manuscript are marked by red color.

**GENERAL COMMENTS**

The article presents a study using seismic data for a minining prospect/site in Norway. The study focuses on applying 3D full waveform inversion and migration to the dataset. To that goal they use a traveltime inversion software and then produce different models and migration images. They conclude that RTM images using FWI models are better than those resulting from constant velocity models or tomographic models. They then proceed to interpret their findings and relate them to prior knowledge of the study area. The main result is a good fit between the mineralisation horizon and a reflector present in the images.

The article is overall well written and the methodologies are clearly explained. The bibliography is sufficient and figures show good quality for publication.

**SPECIFIC COMMENTS**

This is an ambitious exercise that aims at using state-of-the-art imaging algorithms to an onshore dataset. In particular, using FWI for onshore data is difficult and typically results in a lot of trial and error in order to obtain good convergence. The paper is rather clear in explaining the choices made and which ideas resulted in worst results. In any case, several concepts are a bit obscure and may need clarification by the authors. Here follows a list of them.
* * *
**Comment from referee:**

The first obvious question is the choice of algorithms. FWI is an expensive imaging tool. Its use should be justified only of other methodologies fail or are not available. In the manuscript this is not clear. As it is written, the method is a given of the manuscript, but as no novel methodological approaches are presented or benchmarked, some more discussion on the choice of the method would be welcome.

**Author's response:**

We have tried to explain the motivation behind our work in the introduction part [L49-L57], tackling the specific requirement for a velocity model building tool in the hardrock seismic exploration. We are not bringing FWI in our imaging workflow just for its novelty, but it is a real necessity in this case as standard reflection-based methods (such as migration-velocity analysis or reflection tomography) fail as there are no coherent events to drive the velocity model updates. First-arrival tomography (FAT) was successfully used in this context, but it has limited depth penetration in the hardrock environment, as the velocity gradients are typically very low. In our paper, we are exploring to what extent, FWI (in the acoustic version as a first attempt) can improve velocity model required for depth imaging – both in resolution and updates at greater depths.

As a natural consequence of using FWI as a model building tool, we use RTM for the subsequent validation of the velocity models by depth imaging (to rely on the wavefield-based methods, not ray-based methods). In this way, we evaluate a constant velocity model, FAT-derived and FWI-derived velocity models. At this stage, we decided not to make a comprehensive comparison with the time imaging approach of Malehmir et al. (2021, this special issue) or the ray-based depth imaging of Hloušek et al., 2021 (in review for this special issue) (see L524-528).

**Changes in manuscript:** No changes in revised manuscript
* * *
**Comment from referee:**

Furthermore, the authors choose to use acoustic, constant-density FWI for onshore data. This is rather hard to understand, in particular given the lack of long-offset fit that could justify relying on direct P-arrivals and hence benefiting from acoustic FWI. As the results seem to confirm, the inversions mostly affect near-offset structures and might be driven by reflections. In this case, the acoustic approximation fails at reproducing the amplitudes and AVO of data. Some in-depth justification is due in this regard.

**Author's response:**

It is rightly been pointed out that under such conditions, acoustic FWI may not produce the optimum results. Elastic FWI using both diving and reflection modes seems to be the natural choice as we pointed out [L544-L547]. In our case, restrictions such as high-computational costs of modelling and lack of information of physical properties such as S-wave velocity in the study area limited our approach to elastic FWI.

Best to our knowledge, there had been no earlier attempts of building a velocity model using FWI in hardrock environment from surface-seismic data. Therefore, evaluation of the acoustic FWI can be considered as a first-step towards adoption of FWI as a velocity model building tool in crystalline rocks. We used standard techniques used by many researchers to mitigate the elastic effects and AVO trend in the data preprocessing flow [section 3.1.2]. We designed an external mute function to restrict the direct and shear waves, and used data weighting of the misfit function to drive the model updates using far offsets. We treated the data and model as much possible as we can to fulfill the acoustic approximation [L210-211, L255-L259].

**Changes in manuscript:** No changes in revised manuscript
* * *
**Comment from referee:**

Yet another topic not fully covered in depth in this study is the shallow model. In onshore data, local heterogeneities can be large at the first meters. No static corrections seem to be applied in the present case, which seems like a good idea, but one would expect special attention being paid to very near offsets in order to get an approximation of the small-scale shallow model. Ideally, one would invert for surface waves (elastically, that is) or produce an initial model based upon very short offsets. My impression is that the authors ignore these effects and these are "collected" in the wavelet inversion. Such wavelets have quite noticeable amplitude and delay differences with each other. This solution thus potentially averages local effects from both sources and receivers.

**Author's response:**

Thank you for this comment. As you mentioned, the influence of the near-surface heterogeneity on the FWI results / wavelet estimation is obvious. Yet, it seems that there is no universal solution adopted by the researchers investigating FWI on land data on how to handle the near-surface. We followed the most-commonly used approach in which no statics is applied to the data. We have now explicitly mentioned this information in the data preprocessing part in more details, section 3.1.2.

The weathered layer in the hardrock environment is typically 10 to 20 m thick, which can be only approximately resolved using 5-m grid in FAT & 10-m grid in FWI and using only refracted arrivals. The idea of using surface-waves / elastic inversion is great in this regard, but please also note that methods utilizing surface wave information to resolve shallow heterogeneity in crystalline rocks are still

not fully-established, even though some efforts had been recently made e.g., by V. Socco group at Politecnico di Torino. We are aware that these effects can be "collected" in the source inversion, as our accounting for the weathered layer is approximate. We have mentioned this also now explicitly in the revised manuscript. Figure 1 below is comparing source wavelets estimated over the tailing dam and the bedrock, illustrating the effect of the near-surface heterogeneity. We will also mark wavelets estimated from the shots located at the "soft" material in Figure 8. Please also note that the source-to-source and receiver coupling effects were accounted for using surface-consistent amplitude scaling [L217-L218].

**Changes in manuscript:** We have now included the information regarding statics correction and effects related to non-removal of near-surface heterogeneities [L213-217, L262]. We have also modified Figure 8, a red arrow is added to show the shot points located in the vicinity of the tailing dam [L356-357, L383-384].

[Figure]

**Figure 1:** Estimated source wavelet comparison for shots located in the vicinity of tailing dam (bottom left) and bedrock (bottom right). The location of source points for the tailing dam is marked by red circle and the bedrock by blue ellipse. Source wavelets at bedrock characterizes a minimum phase equivalent of the used Vibroseis source while the source signatures at tailing dam are between mixed to zero phase equivalent. Also, the amplitude response is stronger for the bedrock in comparison to tailing dam. The comparison showcases the effect of variable near-surface conditions which are being accumulated in the source wavelet estimation.
* * *
**Comment from referee:**

Something noticeable from Figure 2a is that several anomalies correlate with the acquisition geometry. This might be actual or an artifact of the inversion. Most probably of the regularization used in the FAT. Perhaps the authors could give details in this aspect.

**Author's response:**

Yes, these are the artefacts introduced due to irregular non-standard 3D acquisition [L193-L195]. These shallow anomalies are restricted to a very shallow part of the velocity model and that cannot be removed completely with change of parameterization during the FAT inversion. In order to minimise its effect, we obtained a smoothed version of this model [Figure 2(b,c,d)] after applying a Gaussian filtering both in inline-crossline direction as well as in depth [L195-198, Figure 5a]. We have modified Figure 2 with the incorporation of few depth slices showcasing the change in velocities in the shallow part of the model.

**Changes in manuscript:** We have modified Figure 2 and related text in L186-191.
* * *
**Comment from referee:**

It is unclear throughout the manuscript which norm or cost function is used. Probably it is the L2 norm, the most common in FWI, but this should be clearly stated in the document, together with any specifics used in this respect (e.g. windowing or amplitude normalization).

**Author's response:**

Yes, it is L2 norm. Yes, we applied windowing and trace normalisation [please see L220-222 and Table 1].

**Changes in manuscript:** L137
* * *
**Comment from referee:**

Given the irregular acquisition geometry of the data, I believe that a resolution test would help in determining which parts of the models can we expect to resolve in optimal conditions. It seems to me that some of the deepest parts of the model obtain are being overinterpreted. As we are missing a resolution test, all parts of the obtained model are considered equally resolved and this is misleading.

**Author's response:**

Thank you for pointing this out. However, we prefer not to perform any classical resolution tests like checkerboard tests, which, in our opinion, have limited applicability to FWI. We are aware of the possible artifacts present in the FWI model, yet there seems to be features that are being resolved despite different subsets of shots used in the inversion (as illustrated in Figure 6). This test is also providing us estimate on the parts of the model, which are less reliable. We used different QC method in order to check the reliability of the obtained velocity model i.e. wavelet estimation, data fitting, cost function drop etc. (section 3.1.5). However, we have to stress, that we treat the FWI-derived velocity model in an instrumental way: i.e. the model is used for depth imaging (RTM) and we are making our interpretation based on the RTM images (section 3.2.3). They are confronted with the available geological information [L481-L485]. Of course, our evaluation of different velocity models and corresponding seismic images may seem very subjective and qualitative, but thanks to your suggestion, we have included common-image gathers from RTM to illustrate improvements in gather flatness.

**Changes in manuscript:** No changes in the revised manuscript
* * *
**Comment from referee:**

Another aspect that seems lacking is QC prior to the inversion. Several traces seem prone to cycle skip (e.g. Figs 7 b-d), and a few times throughout the text we are told that long-offset traces cannot be correctly matched. Nevertheless, such traces seem to be part of FWI, which seems like bad practice. Such data cannot help convergence and as such should be removed from FWI. The traces could be used for posterior QC (as in your RMSE visualization) but should be removed from the inversion.

**Author's response:**

Yes, we agree that we could not fully mitigate the local cycle-skipping. But such localized events are prone to areas with poor illumination due to irregular non-standard data acquisition and being in acoustic approximation in an elastic earth. The non-fitting of the far-offset traces (SE end) are attributed to one-way wave propagation (i.e., lack of shots from SE) and FAT's failure to provide kinematically good fit to first arrivals. Although, best to our observation, we obtained a fairly good fit to the first cycle of the waveform for some shots [L342-L350, see Figure 7d for bin centred at 1500 m]. The RMSE maps shows fitting of traces on a numerical scale using the whole waveform. If we compare the two shot gathers in Figure 9, we removed far offsets traces with low S/R on a shot-by-shot basis. Please note that traces with zero value are not used during FWI inversion but are shown in the map. For more clarity, figures are revised by removing all the non-used traces for better visualization.

**Changes in manuscript:** Figure 9 is revised in which non-used traces during FWI are removed [L388-389].
* * *
**Comment from referee:**

Regarding the long offsets, it seems like a lot of computational effort is used in keeping full offsets (i.e. shotgathers are large in the lateral dimensions) for FWI shots but no benefit is obtained from such traces (see Fig 9 for example). In fact, for elastic FWI there is previous work suggesting that removing such offsets can result in better convergence, both in data and model space (see Kormann et al 2017, Comp. Geosciences, for an example). Figure 9 seems to suggest that long offsets do not contribute at all.

**Author's response:**

Yes, it is fair to say that the contribution of far offsets is minimal as compared to short or mid offsets but not zero. As the aim of the study is focused on deeper part of the model and we wanted to take advantage of far offset to drive the updates in the deeper part of the model, hence we kept the receivers from the SE part for more azimuthal coverage and deeper illumination.

**Changes in manuscript:** No changes in the revised manuscript
* * *
**Comment from referee:**

Regarding RTM I have just few concerns. The first is that you seem to keep direct waves in the migration process, which cannot be migrated. You later filter the images to remove the artefacts, but it would be better practice to remove those direct waves from the very beginning.

**Author's response:**

The direct wave / refractions were indeed removed prior to the RTM imaging. We used the same pre-processed dataset as Hloušek et al. (2021) which was used for an advanced-Kirchhoff migration [L398-399].

**Changes in manuscript:** No changes in the revised manuscript
* * *
**Comment from referee:**

Furthermore, in migration we would expect to see some QC in terms of common image gathers or other gathers that can help discern whether the inversion process is correct. This in fact is a QC for the inversion process as well, one of the few QC that can be applied throughout the domain. We expect

gathers from FWI to be flatter than those obtained from FAT or a homogenous model. Some effort in this direction would strongly help improve the confidence of the reader with the results.

**Author's response:**

Yes, you are right that preferably CIG's will be useful in evaluating the quality of the velocity model. On the other hand, there is a significant computational and storage overhead for calculating them. Therefore, we adopted a pragmatic approach in which we validate velocity models by comparing quality of the respective migrated volumes. We followed your suggestion and produced RTM CIGs (surface-offset gathers) using subset of the data (20 shots) for illustration purposes. They are included now as the new Figure 12.

**Changes in manuscript:** New Figure 12 is added in the manuscript. The related text is added as a new section 3.2.4. Also, the later figures are renumbered accordingly in the text and in figure captions.
* * *
**Comment from referee:**

Last but not least, and given the simplicity of the models used, RTM is hard to justify with respect to other cheaper migration techniques, either pre- or post-stack.

**Author's response:**

Recent experience in working with hardrock seismic data (see e.g., references cited in the introduction) is clearly showing the advantage of the pre-stack depth imaging workflows (especially those applied in the shot-gather domain, such as Fresnel Volume Migration, [L41-L48]) over conventional time-domain imaging approach. The choice of the RTM as the imaging "engine" is twofold: first of all, as stated in our reply to point 1, it is directly compatible with the wave-equation based velocity model building offered by FWI. Usually, FWI-derived velocity models need to be smoothed to some degree to be used in ray-based methods. The other issue is that the RTM accounts for all types of waves, including e.g., prism waves, that cannot be easily handled by the ray-based methods. Based on our experience, it looks like RTM is offering certain advantages in imaging, but we don't have enough evidence to claim it is superior to the ray-based pre-stack depth migration algorithms at this stage.

**Changes in manuscript:** No changes in the revised manuscript

**SUGGESTIONS / COMMENTS**

My comments mostly go in the direction that, perhaps, with other decisions in the parameterization and data selection, other results could be obtained, maybe better ones. This does not demerit the results presented, which are interesting and worth reporting, but leaves me wondering if more could be obtained from the data. In the discussion there are some ideas which are interesting, but for the manuscript to feel more complete, some extra effort would be welcome in addressing some of the issues presented above.

My only strong suggestion is a better analysis of the results. Judging the inversion and migration as successful just based on partially better coherence in some reflectors and fit between model and image or image and a single prior structure seems insufficient, given the effort made in producing those models and images. I suggest CIG or alternative methods to compare coherence of the image with respect to offsets or angles at a wide range of locations and depths.

**Author's response:**

We are very much thankful for the in-depth analysis and valuable comments. What we present in our manuscript is already a result of many tests tackling different aspects of FWI, including wavelet inversion, gradient preconditioning, choice of optimization algorithm (L-BFGS vs Steepest Descent), different weighting tests of the misfit function etc. While it may leave an impression that the whole workflow is very subjective, we believe we reached the limits of what the (visco)acoustic FWI can deliver for this kind of data. Because of the aforementioned computational aspects, the CIG's analysis we added in our revised manuscript is limited, but it is showing the improvements in focusing of the image and gather flatness when using FWI-derived velocity model.

**Reviewer 2**

**GENERAL COMMENTS**
* * *
**Comment from referee:**

You should complement the introduction with references to studies from other research groups and not only self-citations, particularly in these sentences outlining general aspects. For example, you could illustrate the examples of use of FWI in hydrocarbon exploration outlined in L62 with apropriate references to different studies (and maybe add other relevant challenges addressed, such as sub-salt or sub-basalt imaging).

**Author's response:**

This point was a bit unclear to us. Regarding the 'general' introduction, it is hard to avoid self-citation aspect, as the hardrock seismic community is relatively small and many members of this community are in fact contributing to this manuscript. Best to our knowledge, this is probably the first application of a joint imaging workflow consisting of FWI and RTM in the hardrock environment. Similarly, we cannot avoid same issue when talking about the depth imaging approach [L48], but we tried to supplement it with other references, for example, Schmelzbach et al. 2008 [L47].

Regarding the industry FWI examples: we talk first about "solving complex imaging challenges, e.g. seeing through gas clouds and resolving shallow velocity heterogeneities" [L58-60], which is mostly relevant to marine environment (similarly with the sub-salt, sub-basalt imaging) and that's why we do not cite any references. Then we turn the reader attention to land data, citing some relevant papers:

"Nevertheless, a few successful case studies have been reported for 2D and 3D land datasets using acoustic/viscoacoustic FWI (Ravaut et al., 2004; Malinowski et al., 2011; Baeten et al., 2013; Adamczyk et al., 2014; Adamczyk et al., 2015; Stopin et al., 2014; Cheng et al., 2017)".

**Changes in manuscript:** The following citations from the original manuscript in L40-41 are removed:

(Milkereit et al., 2000; Malehmir and Bellefleur, 2009; Cheraghi et al., 2012; White et al., 2012; Bellefleur et al., 2015; Koivisto et al., 2015; Yavuz et al., 2015; Bellefleur et al., 2019)

New citations: Malehmir et al., 2012 [L40], Schmelzbach et al., 2008 [L47] and Adamczyk et al., 2015 [L66] are added.
* * *
**Comment from referee:**

I am not a fan of describing the paper organisation in the introduction (last paragraph of the introduction), as it ends up being very redundant (e.g. "Finally, we conclude our case study in the 'Conclusions' section" is quite obvious). The one but the last paragraph looks much better as a wrap up of the introduction section, so I'd suggest removing the last paragraph entirely.

**Author's response:**

Thank you for this suggestion. We have removed the last paragraph from the introduction part.

**Changes in manuscript:** Last paragraph from the original manuscript [L81-88] is removed.
* * *
**Comment from referee:**

L197 - It would be useful to show a map of the raypath coverage, to give a hint of the areas with higher or lower uncertainty. In particular, areas to the SE of the study area do not seem to be sampled by any raypaths; the model in these areas is the initial or an extrapolation from the nearby locations?

**Author's response:**

We have revised Figure 2 where we are now showcasing different depth slices from the tomographic velocity model, masked according to the ray coverage.

**Changes in manuscript:** Figure 2 has been modified. Changes in the text are marked from L186-191.

**SPECIFIC COMMENTS**
* * *
**Comment from referee:**

L34 - "the need for it", explain the "it", use "this technique" or so.

**Author's response:** Suggestion accepted

**Changes in manuscript:** L34-L35
* * *
**Comment from referee:**

L36 – bringS

**Author's response:** Suggestion accepted

**Changes in manuscript:** L36
* * *
**Comment from referee:**

L93 - What does "endowment" mean in this sentence?

**Author's response:** Here we mean the characterisitics of the Bergslagen mineral deposition.

**Changes in manuscript:** No changes in revised manuscript
* * *
**Comment from referee:**

L102 - The references should be ordered either by date of publication (preferrable) or alphabetically, but you have both systems. Please check throughout the manuscript.

**Author's response:** All the references are now ordered by date of publication.

**Changes in manuscript:** L48 and L94.
* * *
**Comment from referee:**

L106 - What other methods are included within that "etc"? Please explain.

**Author's response:** "etc" is removed. Most of the applied methods are already mentioned.

**Changes in manuscript:** L97-98
* * *
**Comment from referee:**

L2014 - "Theoretically, FWI can start with the raw data" - this sentence is too vague, please explain.

**Author's response:** We have modified this statement.

**Changes in manuscript:** L208-209
* * *
**Comment from referee:**

L261 - Does this refer to issues generally observed in land FWI studies or to specific issues observed in your study? If the former, add references; if the latter, please clarify.

**Author's response:** Yes, this a general issue related with land FWI studies. There are not many studies for acoustic-FWI on land data focusing on this particular issue. Therefore, we cannot provide a relevant reference at the moment.

**Changes in manuscript:** No changes in the revised manuscript
* * *
**Comment from referee:**

L444 - You could remove "One can interpret that"

**Author's response:** Suggestion accepted

**Changes in manuscript:** L447

**FIGURES**
* * *
**Comment from referee:**

Figure 10 - Maybe this is a visual effect, but the depth image corresponding to the FWI image (10c) looks like having a higher frequency content than the other two. Is this the case?

**Author's response:** Yes, this is correctly pointed out that the FWI image is having higher frequency content than other two. As RTM is computationally very intense, we decided to run it with the optimal time-step and grid-size based on velocity model used [Line 414, Table 3].

**Changes in manuscript:** No changes in revised manuscript
* * ** * *
**Comment from referee:**

Figure 12 - Please specify in the caption where do the mineralization bodies interpretations come from (otherwise it looks like you are interpreting these bodies from the FWI model).

**Author's response:** Thanks for pointing this out. More information is added.

**Changes in manuscript:** L494

**REFERENCES**
* * *
**Comment from referee:**

Many of the references are incomplete (e.g. lacking th journal name), please check and amend them.

**Author's response:**

Thank you very much for the fine observation. All such discrepancies had been removed in the revised manuscript.

**Changes in manuscript:** Bibliography has been updated accordingly [L590-694].

---

## Author Response (AR2)

First of all, we would like to thank the reviewer for his comments and suggestions. Below we list our detailed answers and changes in manuscript. Lines numbers mentioned here are according to new revised manuscript.

**GENERAL COMMENTS**

This paper addresses a critical topic: how to build effective velocity models in hard rock settings in order to improve the quality of the resultant images. This is exciting to read about, and hopefully is a useful starting point for velocity model building the hard rock world as seismic imaging becomes more frequent. The comments from previous reviewers appeared generally very well thought out and in my opinion the changes that the authors made to address these were valuable, and generally added to the manuscript.

One area where I believe that reviewer #2s comments were perhaps misunderstood was in regards to the introduction, where the reviewer suggested that the authors include additional references by other researchers, but the authors instead removed references that some of them had contributed to. I believe that these references were useful to the reader and suggest they should, in fact, be included in the final draft. The authors should additionally include references to other published FWI efforts in hard rock, such as the paper "Acquisition and Processing of Wider Bandwidth Seismic Data in Crystalline Crust: Progress with the Metal Earth Project" from 2019. Certainly the authors could, at their discretion, also add other papers (by authors unaffiliated with the current work) on 3D seismic undertaken in exploration settings including work by E. Adam, G. Turner and H. Schijns, for example, which I believe was part of the intent of reviewer #2s comment.

**Author's response:** We would like to thank the reviewer for providing their valuable thoughts and pointing out concerns regarding the references. We have accepted the suggestions and updated the citations accordingly (L40-42 and 'references' section accordingly). Regarding the specific citation related to the FWI application in the hardrock environment, best to our knowledge, this is first such attempt in establishing a workflow for building high-resolution velocity model. The suggested publication "Acquisition and Processing of Wider Bandwidth Seismic Data in Crystalline Crust: Progress with the Metal Earth Project" unfortunately only discusses about the low-frequency data acquisition acquired with the aim of velocity model building using FWI which then will be ultimately used during prestack depth migration (PreSDM) to obtain better imaging results. But neither FWI nor PreSDM has been applied or showcased, therefore we think it does not really match with the theme of the article.

**SPECIFIC COMMENTS**
* * *
**Comment from referee:**

1) Amend "shots" and receivers" in Fig 1 and Fig 2 to, respectively, "shots and receivers" and "receivers only" to improve clarity.

**Author's response:**

Suggestion accepted.

**Changes in manuscript:** L127, L205

2) L127 – it would be useful to clarify if this sweep was a linear sweep or low-dwell (or other), as the reader will likely be wondering about the quality of low frequency information available for FWI.

**Author's response:**

Available information has been added in the revised manuscript.

**Changes in manuscript:** L119

3) L198 – it would be helpful to include the elevation range of the receivers/shots within the survey as the depth of the slices shown subsequently is not currently clear.

**Author's response:**

Suggestion accepted.

**Changes in manuscript:** L207-208.

4) L285 – the FWI was run from 6-25 Hz, but the sweep started at 10 Hz, and some geophones had a 28 Hz frequency. It would be useful for the authors to comment on how much information was actually present in the data at these lower frequencies.

**Author's response:** Additional information has been added in the revised manuscript.

**Changes in manuscript:** L240-242

5) In the discussion it would be helpful to address any observations the authors may have made on ideal acquisition parameters, to inform future trials. Eg. Would the authors have used lower frequency geophones or a lower starting frequency on the sweep in future work? Were there any issues with blending different receivers in the inversion?

**Author's response:** Thank you very much for pointing out this. We have tried to cover this point already in the manuscript in terms of ideal acquisitions setups for future and other technical details that we would like to be incorporated in future studies. We have now incorporated more details specific to hardrock environment, please see L549-553.

**Additional change in manuscript:** Figure 8 (it has figure captions missing)